# Continuous-wave upconversion lasing with a sub-10 W cm$^{-2}$ threshold enabled by atomic disorder in the host matrix

Byeong-Seok Moon [1], Tae Kyung Lee [2,3], Woo Cheol Jeon[2], Sang Kyu Kwak [2], Young-Jin Kim[4✉] & Dong-Hwan Kim [1,5✉]

Microscale lasers efficiently deliver coherent photons into small volumes for intracellular biosensors and all-photonic microprocessors. Such technologies have given rise to a compelling pursuit of ever-smaller and ever-more-efficient microlasers. Upconversion microlasers have great potential owing to their large anti-Stokes shifts but have lagged behind other microlasers due to their high pump power requirement for population inversion of multiphoton-excited states. Here, we demonstrate continuous-wave upconversion lasing at an ultralow lasing threshold (4.7 W cm$^{-2}$) by adopting monolithic whispering-gallery-mode microspheres synthesized by laser-induced liquefaction of upconversion nanoparticles and subsequent rapid quenching ("liquid-quenching"). Liquid-quenching completely integrates upconversion nanoparticles to provide high pump-to-gain interaction with low intracavity losses for efficient lasing. Atomic-scale disorder in the liquid-quenched host matrix suppresses phonon-assisted energy back transfer to achieve efficient population inversion. Narrow laser lines were spectrally tuned by up to 3.56 nm by injection pump power and operation temperature adjustments. Our low-threshold, wavelength-tunable, and continuous-wave upconversion microlaser with a narrow linewidth represents the anti-Stokes-shift microlaser that is competitive against state-of-the-art Stokes-shift microlasers, which paves the way for high-resolution atomic spectroscopy, biomedical quantitative phase imaging, and high-speed optical communication via wavelength-division-multiplexing.

[1] School of Chemical Engineering, Sungkyunkwan University, Suwon, Republic of Korea. [2] Department of Energy Engineering, School of Energy and Chemical Engineering, Ulsan National Institute of Science and Technology (UNIST), Ulsan, Republic of Korea. [3] Photovoltaics Research Department, Korea Institute of Energy Research (KIER), Daejeon, Republic of Korea. [4] Department of Mechanical Engineering, Korea Advanced Institute of Science and Technology (KAIST), Daejeon, Republic of Korea. [5] Biomedical Institute for Convergence at SKKU (BICS), Sungkyunkwan University, Suwon, Republic of Korea. ✉email: yj. kim@kaist.ac.kr; dhkim1@skku.edu

Microscale lasers have become crucial to widespread applications of biophotonics[1–3] and on-chip optoelectronics[4–6] for the efficient delivery of coherent photons into tiny target volumes with precisely controlled wavelengths[5]. Such demands have given rise to a compelling pursuit of smaller but more efficient microlasers. In particular, a low-threshold, wavelength-tunable, and continuous-wave microlaser with a narrow linewidth has been sought for realizing high-resolution spectroscopy[7–9], interferometric phase imaging[10,11], and wavelength-division-multiplexing[12,13] in a microscale device. Despite the great progress of microlasers, these applications have remained challenging because miniaturization of the optical cavity makes lasing difficult due to high intracavity loss[14,15] and insufficient pump-to-gain interaction[16,17].

Lanthanide-doped upconversion nanoparticles (UCNPs) enable large anti-Stokes-shift emission, which generates a high-energy photon (i.e., ultraviolet or visible light) via multiphoton absorption of low-energy photons (i.e., near-infrared light)[18]. In addition, compared to traditional luminophores such as fluorescent dyes and quantum dots, UCNPs are photobleaching-resistant, nonblinking, photostable, and have real intermediate energy states for efficient population inversion of multiphoton-excited states[19,20]. Such features of UCNPs make them an attractive gain medium for the development of anti-Stokes-shift microlasers, which provide deeper penetration depth, higher signal-to-noise ratio, and less damage to devices than conventional Stokes-shift microlasers[21–23]. Therefore, for the past decade, a number of attempts have been made to realize anti-Stokes-shift microlasers using UCNPs, mainly by coating UCNPs on the surfaces of spherical or cylindrical dielectric-cavity-based microresonators[22,24–27]. These microlasers typically involve a lasing threshold ($>10^2$ W cm$^{-2}$) that is two orders of magnitude higher than that associated with Stokes-shift laser operation ($<10^1$ W cm$^{-2}$)[6,28] due to the high pump power requirement for multiphoton absorption. Most recently, UCNP-coated plasmonic nanopillar arrays realized continuous-wave upconversion lasing at 29 W cm$^{-2}$ by the aid of lattice plasmons[21], but the laser linewidth was fairly broad (~1 nm) due to propagation loss of the nanoscale metallic cavity[6]. Despite the significant progress in upconversion small lasers, the direct use of UCNPs as a gain medium inevitably causes scattering loss of photons at the interfaces, small pump-to-gain interaction limited to the thin outer layer, and a risk of degradation during long-term operation, which need to be overcome for realization of practical applications.

In this work, we describe the realization of a sub-10 W cm$^{-2}$ threshold continuous-wave upconversion microlaser by rapid quenching of laser-molten UCNPs, which is called the liquid-quenching method[29,30]. Liquid quenching facilitates complete integration of UCNPs into a monolithic microsphere as small as 2.44 μm in diameter with a highly smooth surface texture, resulting in lower intracavity loss with high pump-to-gain interaction for efficient light coupling to whispering-gallery-mode (WGM) resonators, in contrast to previous upconversion microlasers[22,24–26]. From the point of view of gain materials, we newly found that the highly disordered microenvironments of the liquid-quenched amorphous matrix effectively suppress phonon-assisted energy back transfer (EBT) from the activator (i.e., Er$^{3+}$ and Tm$^{3+}$) to the sensitizer (i.e., Yb$^{3+}$), which has hindered population inversion in the presence of the sensitizer. Overall, these superior characteristics of our liquid-quenched upconversion microspheres (LQUMs) resulted in an ultralow upconversion lasing threshold of 4.7 W cm$^{-2}$ with a linewidth as narrow as 0.27 nm, which makes the anti-Stokes-shift-based microlaser competitive against the state-of-the-art Stokes-shift microlasers[6,28]. Our upconversion microlasers provide a high wall-plug pump efficiency because of direct pumping by 980-nm near-infrared diode lasers, which are widely used in laboratories and industry. The LQUM microlaser was also operated with alternative doping ions, implying the device's universal applicability with other lanthanides for the expansion of output laser wavelengths. Last, the laser output wavelengths of our upconversion microlaser are readily tuned to be as large as 3.56 nm by thermally expanding the cavity size, which was controlled by changing the injection pump power and operation temperature. This tuning process over a wide spectral range was much simpler than that for conventional semiconductor lasers[31].

## Results and discussion

Monolithic lanthanide-doped microspheres, serving as microscale WGM resonators, were fabricated by liquid quenching of UCNPs as shown in Fig. 1a following the procedures described in our previous report[30] (see Supplementary Fig. 1 for details). A group of silica-coated β-NaYF$_4$:Yb$^{3+}$, Er$^{3+}$ (20%, 2%) UCNPs on a TEM grid was liquefied by a highly focused continuous-wave 980-nm laser beam with a photon density of 3.0 MW cm$^{-2}$. The molten upconversion host matrix was irreversibly solidified into a uniform and smooth microsphere by surface tension during liquefaction as shown in Fig. 1b (the microstructural analysis of the liquid-quenched amorphous materials is provided in our previous report[30]). Because the surface tension results in excellent surface finish[32], noticeable defects for the scattering center were not observed on the microsphere's surface in a high-resolution (1.4 nm) SEM image. As a result, the fabricated LQUM served as an excellent optical resonator that features strong light confinement in WGMs for laser operation[33] (Fig. 1c).

An LQUM of a 2.44 μm in diameter (Fig. 1b) was investigated for continuous-wave upconversion lasing by pumping with a 980-nm diode laser at a power density of 3.16 MW cm$^{-2}$ (Fig. 1d). Because the upconversion lasing and upconversion luminescence occur simultaneously at the pumping spot, we observed the upconversion lasing at the other side away from the pumping spot to accurately investigate the lasing characteristics. The small and smooth WGM resonator exhibited excellent laser emission characteristics. The intense laser outputs were observed at multiple emission bands of 525, 550, and 665 nm. The narrowest spectral linewidth was measured to be 0.27 nm at 525.79 nm, where the Q factor was as high as 1947, and the free spectral range corresponded to 22.44 nm; the Q factor implies a low optical loss in the resonator, and the free spectral range reflects the small size of the resonator[33]. The average output power of each lasing mode was stable without significant drift or fluctuation (Supplementary Fig. 2) over the entire investigation period of several days. Additionally, the laser emission exhibited negligible sensitivity to the pump laser's polarization axis owing to the amorphous nature of the LQUM[30] (Supplementary Fig. 3), which relaxes pump requirements.

We also investigated the size effect of the LQUM on the laser emission characteristics because WGM resonances are highly sensitive to the resonator's size[33]. We found that the best laser emission was observed from an LQUM of 2.44 μm in diameter (Fig. 1b, d). Note that the LQUM larger than the laser spot size (2 μm × 4 μm, an elliptical shape originated from the astigmatism of diode lasers) is hardly fabricated because UCNPs out of the laser spot cannot participate in the liquid-quenching process. At diameters smaller than 2.44 μm, an exponential increase in curvature loss[1] and a resonance mismatch for each emission band impedes efficient laser emission in the green and red regions (Supplementary Fig. 4). Nevertheless, the proposed liquid-quenching method allows us to miniaturize the upconversion microlaser down to the smallest size ever[22].

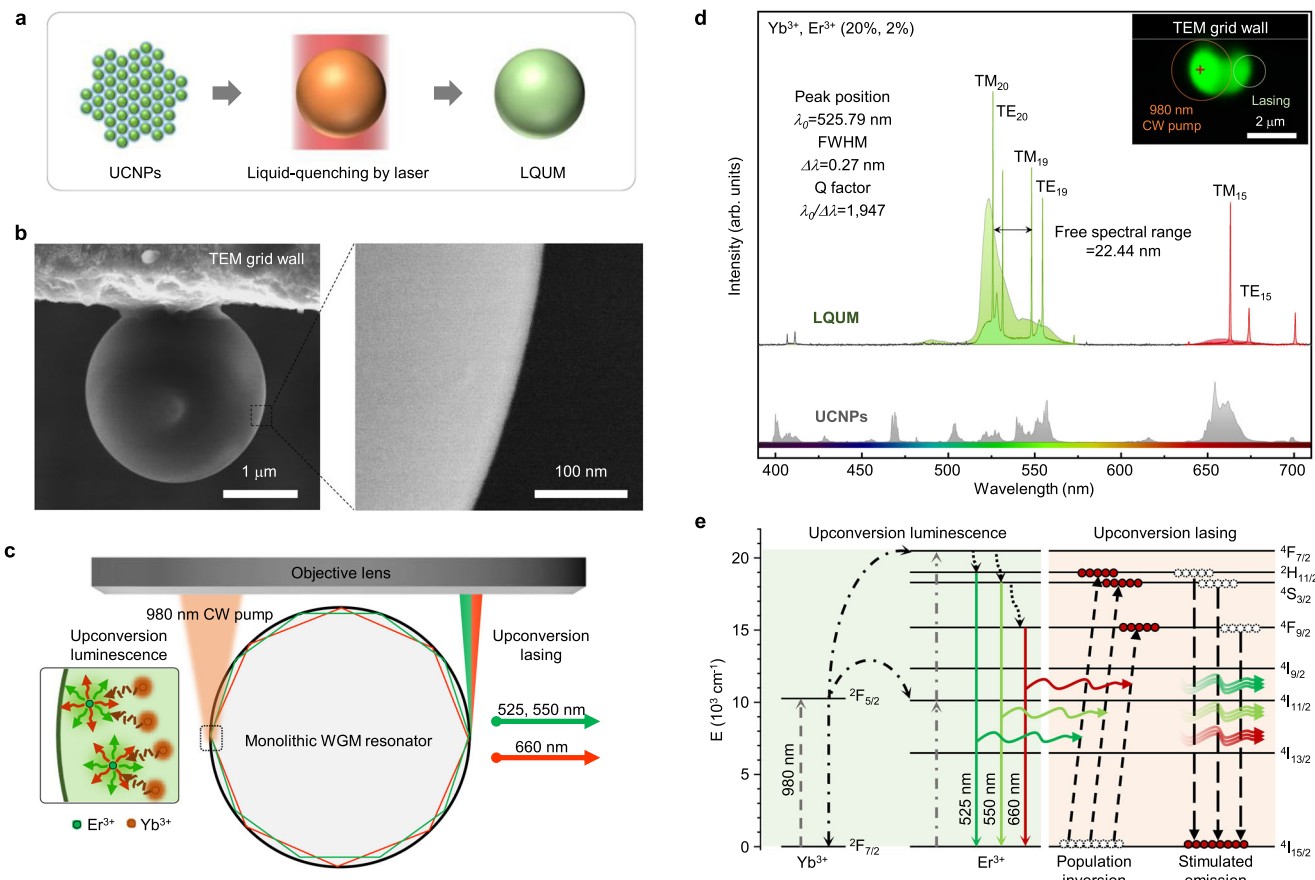

**Fig. 1 Continuous-wave upconversion lasing in liquid-quenched upconversion microspheres (LQUM). a** Schematic of the fabrication of an LQUM via the liquid quenching of silica-coated β-NaYF$_4$:Yb$^{3+}$, Er$^{3+}$ (20%, 2%) UCNPs on a TEM grid using a highly focused CW 980-nm laser beam. **b** SEM images of the LQUM at low and high magnifications. Because the direction of light circulation must be aligned perfectly with the circumference of the microsphere, an LQUM dangling on a TEM grid wall was selected for the formation of WGM resonators. **c** Schematic of upconversion lasing in an LQUM using a free-space beam of a 980-nm continuous-wave pump laser. The LQUM accommodates upconversion lasing through the light circulation along the circumference of the microsphere upon the upconversion luminescence facilitated by energy transfer upconversion of an activator (Er$^{3+}$) with a sensitizer (Yb$^{3+}$). **d** Upconversion luminescence of UCNPs (below) and an LQUM with upconversion lasing (above). The doping concentrations of Yb$^{3+}$ and Er$^{3+}$ were 20% and 2%, respectively. The color inset shows an optical microscopy image of an illuminated LQUM. **e** The proposed upconversion lasing pathway supported by energy transfer upconversion in the LQUM. Short-dashed, dot-dashed, colored solid, long-dashed, and dotted lines indicate absorption, energy transfer, spontaneous emission, stimulated emission, and multiphonon relaxation, respectively.

Energy transfer upconversion (ETU) is the most efficient upconversion mechanism that utilizes a sensitizer (e.g., Yb$^{3+}$) to enhance the upconversion efficiency[34]. Nevertheless, because the use of sensitizers generally hinders the population inversion of activators (e.g., Er$^{3+}$ or Tm$^{3+}$) through detrimental EBT to sensitizers, ETU has been difficult to incorporate into continuous-wave upconversion microlasers[18]. However, unlike conventional upconversion materials, LQUMs provide a unique ETU pathway that effectively dominates over deleterious EBT; the upconverted photons in an LQUM can be effectively concentrated in the upconversion lasing states of $^2H_{11/2}$, $^4S_{3/2}$ ($\rightarrow ^4I_{15/2}$), and $^4F_{9/2}$ ($\rightarrow ^4I_{15/2}$) for green (525 nm, 550 nm) and red (660 nm) bands, respectively (Fig. 1e). This unique advantage in ETU for achieving efficient population inversion in LQUMs will be further discussed using atomic-scale simulations.

Because of the seamless cavity design and high surface smoothness with internally incorporated lanthanides, LQUMs provide higher pump-to-gain interaction and lower intracavity loss than conventional upconversion microlasers[22,24–26]. To exploit this advantage, we carefully investigated the optimal pumping position from the very edge to the center of an LQUM and found that the optimal pumping position for the lowest

lasing threshold is located at the rim of the LQUM (~21% of the radius far from the surface of the microsphere) (the experimental and theoretical investigations on the pump position effects are described in Supplementary Note 1 and Supplementary Figs. 5–8).

After the pumping position optimization, continuous-wave upconversion lasing was successfully generated at pump power densities below $10^3$ W cm$^{-2}$ (Fig. 2a, b). In particular, the lowest lasing threshold of 4.7 W cm$^{-2}$ with significant linewidth narrowing was observed at $^4F_{9/2} \rightarrow ^4I_{15/2}$ (Fig. 2c, d), which, to the best of our knowledge, is the lowest threshold ever reported for anti-Stokes-shift microlasers. The laser emission bands were generated following the ascending order of the energy levels of the excited states for each band, from $^4F_{9/2}$, $^4S_{3/2}$ to $^2H_{11/2}$, as shown in Fig. 2d, e. This order indicates that multiphonon relaxation is involved in the lasing process to populate the lower-energy excited states (see the details in Fig. 3f). The lasing thresholds of each emission band were determined by the slope change in a log–log plot of the emission intensity depending on pump power density, which represents the transition of the emission behavior of Er$^{3+}$ from spontaneous emission to stimulated emission[25]. The higher slope values of green emission bands ($^4F_{9/2}$, $^4S_{3/2} \rightarrow ^4I_{15/2}$)

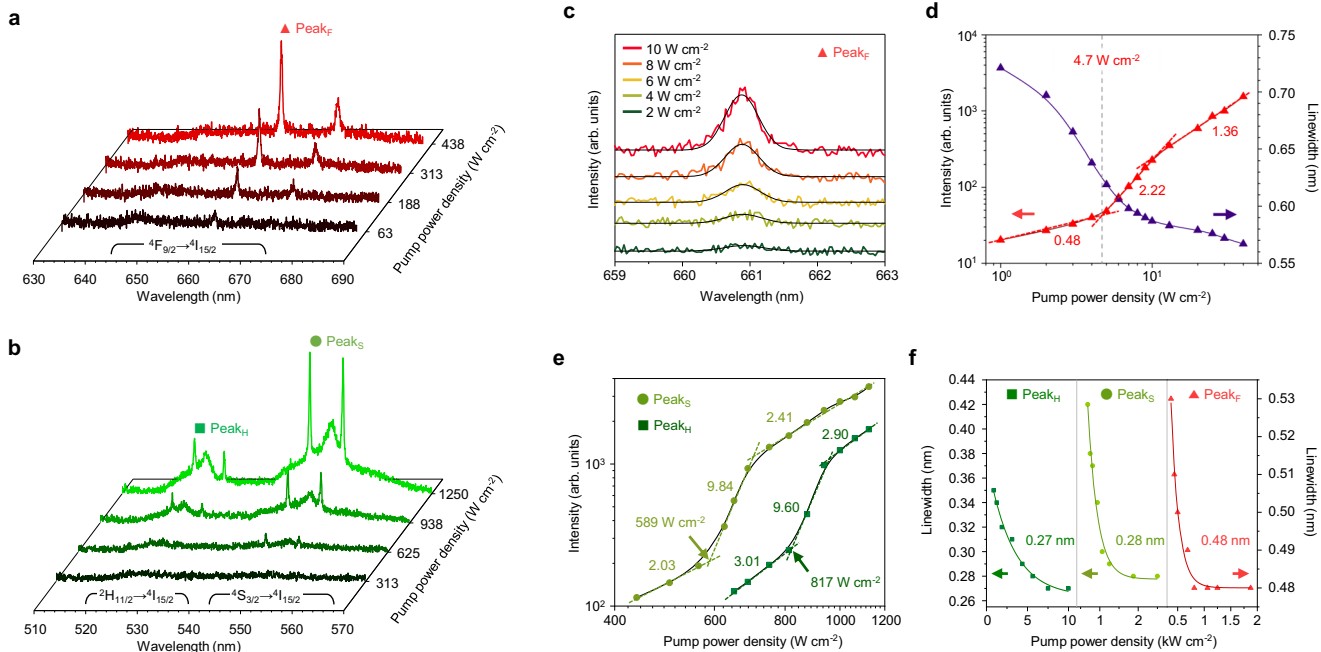

**Fig. 2 Ultralow-threshold continuous-wave upconversion lasing. a, b.** Red- (**a**) and green-band (**b**) emission spectra of an LQUM doped with $Yb^{3+}$ and $Er^{3+}$ around the lasing threshold. The laser lines corresponding to $^2H_{11/2} \rightarrow {}^4I_{15/2}$, $^4S_{3/2} \rightarrow {}^4I_{15/2}$ and $^4F_{9/2} \rightarrow {}^4I_{15/2}$ are labeled as $Peak_H$, $Peak_S$, and $Peak_F$, respectively. **c, d** Laser emission spectrum of $Peak_F$ near the laser threshold of sub-10 W $cm^{-2}$ (**c**) and a laser emission intensity and linewidth measurement (**d**). **e** Generation of $Peak_H$ and $Peak_S$. **f** Linewidth saturation of the laser lines with increasing pump power density.

than red emission band ($^4F_{9/2} \rightarrow {}^4I_{15/2}$) at lasing thresholds might be attributed to the higher amount of input photons at green emission bands than red emission band (see the upconversion luminescence of the LQUM that has intense single-band emission at green region as shown in Fig. 1d). As the pump power increases, the intensity of the laser lines increases proportionally to the upconversion luminescence intensity of the corresponding emission bands, where the power dependence of the green emission band is higher than that of the red emission band (Supplementary Fig. 9), thus finally yielding an intense laser spectrum, as shown in Fig. 1d. Above the lasing threshold, the spectral linewidth of each lasing mode continuously decreases and converges to 0.27, 0.28, and 0.48 nm at 525, 550, and 665 nm, respectively (Fig. 2f). This continuous linewidth narrowing, after the lasing threshold, along with the increasing pump power can be attributed to the increase of the degree of coherence of the stimulated emission because the number of the coherent photons that traveling around the cavity become larger at higher pump power[35,36].

To determine the origin of the ultralow lasing thresholds of the LQUM, we investigated the ETU pathway involved in the upconversion lasing. Because energy transfers of $Ln^{3+}$ accompany phonon production or annihilation to make up for the energy mismatch between different excited states[37], we analyzed the phonon properties of the LQUM through atomistic simulations. We previously reported that the complete amorphization of UCNPs was realized by incorporating $SiO_2$ into a hexagonal $NaYF_4$ matrix through liquid quenching[30]. By mimicking the experimental procedure of liquid quenching, we built amorphous model systems using an all-atom molecular dynamics (AAMD) simulation followed by an ab initio molecular dynamics (AIMD) simulation (Fig. 3a, see computational details in Supplementary Note 2) and found that the liquid-quenched amorphous material provides an atomically disordered host matrix for $Ln^{3+}$ (i.e., all the atomic bonds in the first coordination sphere of $Ln^{3+}$ are non-centrosymmetric), unlike the conventional crystalline and

glass upconversion host materials, due to the rapid quenching of randomly mixed elements in the liquid phase (the atomic configurations of the liquid-quenched amorphous phase and the hexagonal crystal phase for simulations are depicted in Supplementary Figs. 10–15).

From density functional theory (DFT) calculations, we found that the phonon density of states (DOS) of $Yb^{3+}$ and $Er^{3+}$ were noticeably decreased in the amorphous phased system after liquid quenching regardless of the atomic configuration of the amorphous phase compared to the hexagonal crystal phase (Fig. 3b and Supplementary Fig. 16 for $Yb^{3+}$ and Fig. 3c and Supplementary Fig. 17 for $Er^{3+}$). This phenomenon can be attributed to the fact that the compositional and structural disorders in the lattices restrict the available phonon modes owing to the tight localization of phonons[38,39].

The decreased phonon DOS of $Er^{3+}$ and $Yb^{3+}$ in the LQUM can significantly affect the probability of EBT. Based on the Miyakawa and Dexter theory, the probability of phonon-assisted energy transfer ($P_{PET}$) can be expressed by the following equation[40]:

$$P_{PET} \approx e^{-\alpha \Delta E} \tag{1}$$

where $\alpha$ is a physical parameter inversely proportional to the phonon DOS intensity of the sensitizer and activator in the host matrix. $\Delta E$ is the energy mismatch between the energy levels of lanthanide ions where EBT occurs by phonon-assisted energy transfer (Fig. 3e). This equation indicates that our amorphous system can efficiently reduce the phonon-assisted EBT due to a significant decrease in the phonon DOS intensities of $Yb^{3+}$ and $Er^{3+}$. Note that ETU from $Yb^{3+}$ to $Er^{3+}$, which is the main upconversion mechanism, has been realized without phonon assistance because of the large spectral overlap between $^2F_{7/2} \rightarrow {}^2F_{5/2}$ of $Yb^{3+}$ and $^4I_{15/2} \rightarrow {}^4I_{11/2}$ and $^4I_{11/2} \rightarrow {}^4F_{7/2}$ of $Er^{3+}$ (ref. [41]).

Then, we investigated the multiphonon relaxation rate of $Er^{3+}$, which is responsible for the population of the upconversion

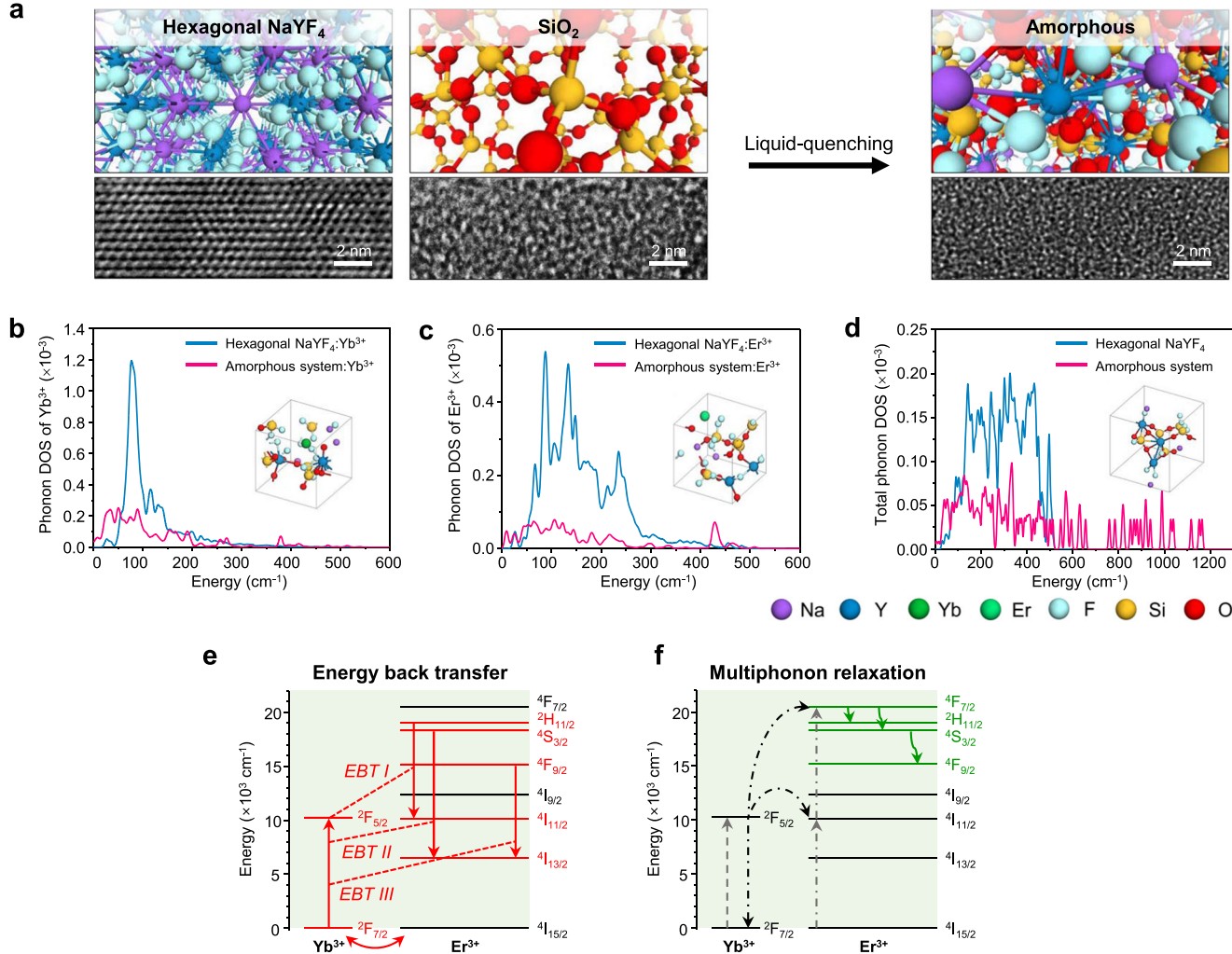

**Fig. 3 Theoretical analysis of the enhancement of the upconversion lasing efficiency in the amorphous phase of NaYF$_4$ combined with SiO$_2$. a** Schematic and TEM images of the microstructure of the amorphous phase after liquid quenching of hexagonal NaYF$_4$ with SiO$_2$. **b** Phonon DOS of Yb$^{3+}$ in the Yb-doped amorphous system (Model 6 in Supplementary Fig. 12, which shows the largest difference from hexagonal crystal NaYF$_4$) and Yb-doped hexagonal crystal NaYF$_4$ (Supplementary Fig. 15b). **c** Phonon DOS of Er$^{3+}$ in the Er-doped amorphous system (Model 5 in Supplementary Fig. 13, which shows the largest difference from hexagonal crystal NaYF$_4$) and Er-doped hexagonal crystal NaYF$_4$ (Supplementary Fig. 15c). **d** Total phonon DOS of the amorphous system (Model 2 in Supplementary Fig. 11) and hexagonal crystal NaYF$_4$ (Supplementary Fig. 15a). Note that the total phonon DOS are normalized by dividing by the total number of atoms in each system. **e, f** Mechanisms of energy back transfer (Er$^{3+}$ → Yb$^{3+}$) (**e**) and multiphonon relaxation (Er$^{3+}$) (**f**). ΔE for energy back transfer: ~1373.4 cm$^{-1}$ for EBT I (between $^2F_{7/2}$ → $^2F_{5/2}$ (Yb$^{3+}$) and $^2H_{11/2}$ → $^4I_{11/2}$ (Er$^{3+}$)), ~1553.1 cm$^{-1}$ for EBT II (between $^2F_{7/2}$ → $^2F_{5/2}$ (Yb$^{3+}$) and $^4S_{3/2}$ → $^4I_{13/2}$ (Er$^{3+}$)) and ~1563.7 cm$^{-1}$ for EBT III (between $^2F_{7/2}$ → $^2F_{5/2}$ (Yb$^{3+}$) and $^4F_{9/2}$ → $^4I_{13/2}$ (Er$^{3+}$)). ΔE for multiphonon relaxation of the Er$^{3+}$ ion: $^4F_{7/2}$ → $^2H_{11/2}$ = ~1483 cm$^{-1}$, $^4F_{7/2}$ → $^4S_{3/2}$: ~2195 cm$^{-1}$, and $^4S_{3/2}$ → $^4F_{9/2}$ = ~3117 cm$^{-1}$. Note that red solid arrows with red dashed lines indicate energy transfers by EBT. Red curved arrows indicate an interaction between Yb$^{3+}$ and Er$^{3+}$. Gray dashed, black dot-dashed, and green solid arrows indicate absorption, energy transfer, and multiphonon relaxation, respectively.

emission states $^2H_{11/2}$, $^4S_{3/2}$, and $^4F_{9/2}$ in our amorphous system. Generally, the multiphonon relaxation rate ($P_{MR}$) can be explained by van Dijk's modified energy gap law with the following equation[42]:

$$P_{MR} \approx e^{(2\hbar\omega_{max} - \Delta E)} \qquad (2)$$

where $\hbar\omega_{max}$ is the highest phonon energy of the host matrix. ΔE is the energy gap between the energy levels where multiphonon relaxation occurs (Fig. 3f). Because the multiphonon relaxation rate is affected by the highest phonon energy of the host matrix[42], we calculated the total phonon DOS of our amorphous host matrix. We found that the amorphous systems showed higher phonon DOS at 600 cm$^{-1}$ ~ 1250 cm$^{-1}$ than the hexagonal crystal NaYF$_4$ (Fig. 3d and Supplementary Fig. 18) and were mainly contributed by fluorine, silicon, and oxygen atoms, probably due to the high phonon energy involved with Si–O and

Si–F bonds[43] (Supplementary Fig. 19). Accordingly, the multi-phonon relaxation rate in the Er$^{3+}$ activator of the amorphous system is higher than that of the conventional hexagonal crystal NaYF$_4$. Note that the simulated phonon energy of the LQUM is supported by the greatly reduced Raman emission intensity of the LQUM compared with β-NaYF$_4$ (Supplementary Fig. 20). These unique features, both the suppression of EBT from the Er$^{3+}$ activator to the Yb$^{3+}$ sensitizer and the allowed multiphonon relaxation in the Er$^{3+}$ activator, facilitate efficient ETU for the population inversion at the upconversion emission states without significant energy loss (Fig. 1e), which results in the ultralow-threshold upconversion lasing of the LQUM microlasers.

Next, we analyzed the spatial, spectral, and polarization characteristics of the LQUM microlaser in accordance with classical laser physics, as shown in Fig. 4. The upconversion lasing outputs are emitted from both sides of the microsphere edge; one

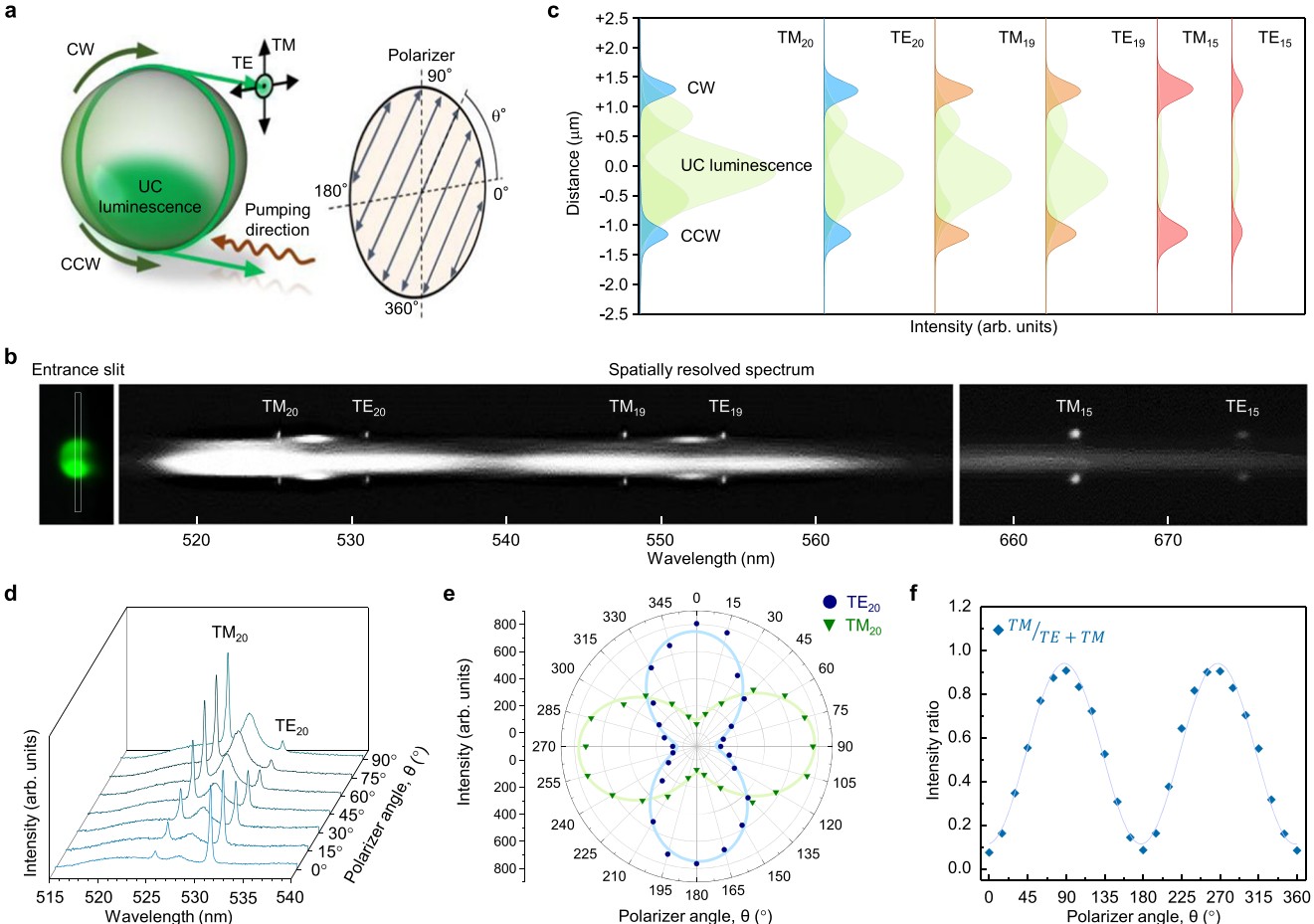

**Fig. 4 Spatial, spectral, and polarization analysis of laser emission. a** Schematic of WGM analysis in an LQUM determined by circulation direction (clockwise (CW) and counterclockwise (CCW)) and transverse modes (transverse magnetic (TM) and transverse electric (TE) modes). **b** Spatially resolved spectrum of laser emission from the LQUM. The narrow rectangular section was spatially selected by the spectrograph's entrance slit of 2 μm and resolved using a high-resolution grating of 1200 lines mm⁻¹ (centered at 500 nm). Along with vertical direction, each point of the selected area generates emission spectrum horizontally on the panel of EMCCD to create the image of **b**. **c** Deconvolution of the laser lines using Gaussian-Lorentz fitting. **d–f** Polarization investigation of the laser lines using the emission spectrum (**d**), a polar plot of the intensities (**e**) and the intensity ratio (**f**). The fitting curves in **e** and **f** were drawn by a cosine-square function.

circulates in the clockwise direction along the pumping direction, whereas the other circulates in the counterclockwise direction[44]. Each laser line showed a distinct polarization mode, transverse electric (TE) mode, or transverse magnetic (TM) mode; the overall lasing wavelengths from mode numbers 15 to 20 agree well with the theoretical calculation based on asymptotic solutions[45] (see Supplementary Fig. 21). The finite element method (FEM)-based simulations were also supported the WGM of LQUM (Supplementary Note 3 and Supplementary Fig. 22). The emission of the LQUM all exhibited well-defined Gaussian shapes for both the TE and TM modes over the broad spectral range from 520 to 680 nm (Fig. 4b, c); this finding was confirmed by deconvoluting the spatial modes into central broadband upconversion luminescence and narrow-band side modes (Fig. 4c). The same spatial profile and spectral distribution of the pair of emission with different circulating directions verify the high-quality surface and high internal homogeneity of the microspheres[46]. The emission polarization states of the TE and TM modes were confirmed by rotating a birefringent polarizer with the high extinction ratio of 10⁶:1, as shown in Fig. 4a. Each lasing mode showed a distinct linear TE or TM polarization state (Fig. 4d, e), which agreed well with the theoretical expectation for the WGM. For the TM₂₀ mode, the intensity ratio between the two polarization states was 90:10, as shown in Fig. 4f.

The wavelength coverage of the LQUM laser emission could be conveniently changed from the green/red emission bands to the blue emission band by simply substituting $Er^{3+}$ with $Tm^{3+}$ because, as we confirmed in Figs. 2 and 3, the light amplification of the LQUM was realized without a dopant-specific upconversion mechanism that could restrict the choice of activator ions[22]. Our DFT calculations also confirmed that the phonon DOS of $Tm^{3+}$ was decreased in the amorphous system compared with the crystal system (Supplementary Fig. 23). We observed robust continuous-wave upconversion lasing in the blue emission bands in the $Tm^{3+}$-doped LQUM that was 1.99 μm in diameter, which supports WGM resonances at ~475 nm; the lasing threshold was as low as 1.5 kW cm⁻², ~30 times lower than the previous record[22] (Fig. 5).

Wavelength tunability is a prerequisite for high-resolution spectroscopies that vary the wavelength of a continuous-wave laser to scan over atomic/molecular absorption lines and for on-chip optical signal processors that require a large number of signal channels for dense wavelength-division-multiplexing. The output wavelengths of our upconversion microlasers can be tuned either by controlling the WGM cavity size via the surrounding temperature or by adjusting the injection pump power density (Fig. 6a). The output wavelengths were red-shifted by ~2.62 nm by increasing the pump power density from 0.01 to 3.16 MW cm⁻² at

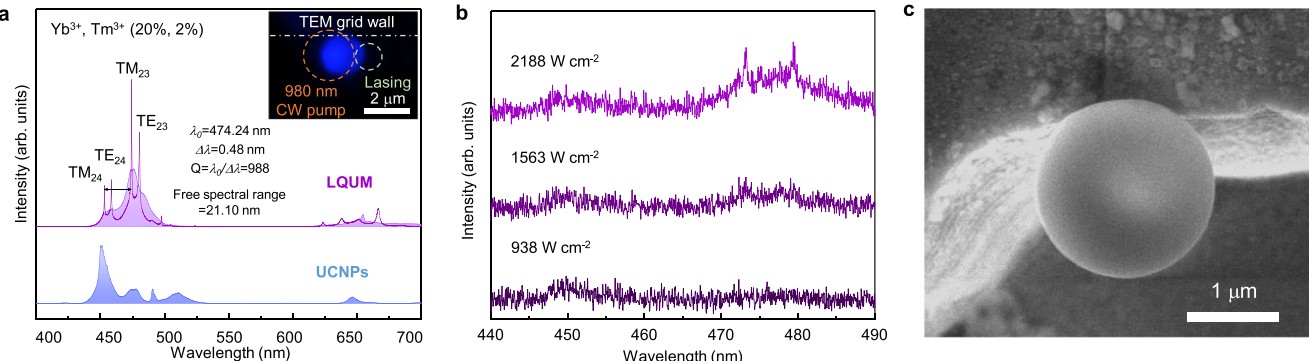

**Fig. 5 Upconversion lasing with Tm³⁺.** **a** Upconversion luminescence and upconversion lasing in an LQUM doped with Yb³⁺ and Tm³⁺ (20 and 2%). The color inset shows an optical microscopy image of an illuminated LQUM. **b** Emission spectrum in the blue emission band around the lasing threshold of 1.5 kW cm⁻². **c** A SEM image of an LQUM measuring 1.99 μm in diameter doped with Yb³⁺ and Tm³⁺.

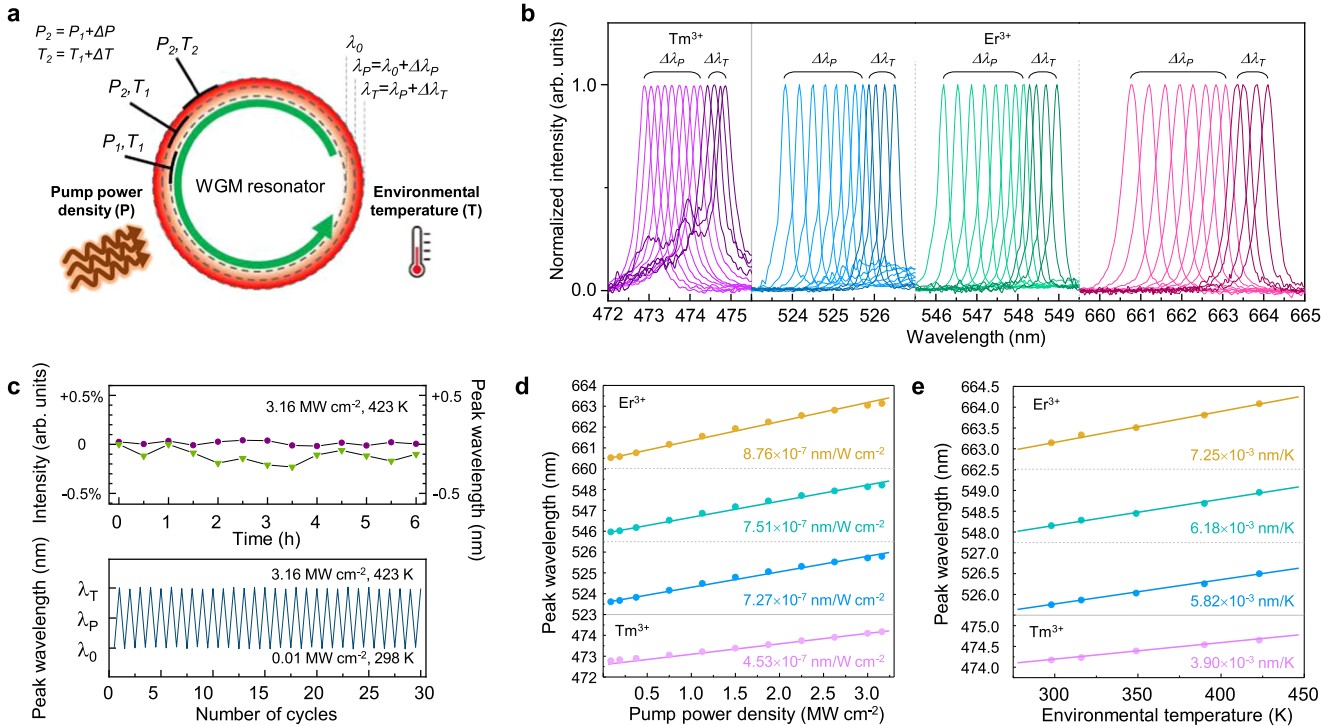

**Fig. 6 Tuning of laser emission wavelengths.** **a** Schematic of wavelength tuning achieved by controlling the pump power density (ΔP) and operation temperature (ΔT). The initial lasing wavelength and the shifted lasing wavelengths under the control of the pump power density and operation temperature are indicated by $\lambda_O$, $\lambda_P$ and $\lambda_T$, respectively. **b** Emission spectra of spectrally tuned lasers. $\Delta\lambda_P$ and $\Delta\lambda_T$ indicate shifts in the lasing wavelength with increasing pump power density (from 0.01 to 3.16 MW cm⁻²) and operation temperature (from 298 to 423 K). **c** Stability analysis in terms of static (above) and dynamic (below) operations. Green and purple lines in static operation indicate the lasing peak intensity and spectral position depending on time, respectively. **d**, **e** Linear plots of the laser line peak wavelengths as a function of pump power density (**d**) and operation temperature (**e**).

298 K and then red-shifted further by ~0.94 nm by increasing the operation temperature of the LQUM microlaser from 298 to 423 K (Fig. 6b) using a home-built microscopic temperature controller (Supplementary Fig. 24). The increase in the internal temperature of the LQUM microlaser causes thermal expansion of the cavity which induces a red-shift of the lasing wavelengths[47,48].

Owing to the excellent photo, thermal, and structural stabilities inherited from inorganic upconversion host materials[49], the lasing was well sustained even in a harsh operation environment—at a power density of 3.16 MW cm⁻² and a temperature of 423 K—without noticeable degradation over 6 h of continuous operation (Fig. 6c, above). The wavelength tuning was confirmed to be repeatable over 30 cycles in a fully reversible manner (Fig. 6c, below). The laser wavelengths were linearly proportional to the

applied pump power density (Fig. 6d) and the operation temperature (Fig. 6e). Based on the fact that the lasing wavelength shift is proportional to the initial peak emission wavelength, the largest shift of 3.56 nm with the highest sensitivity of $8.76 \times 10^{-7}$ nm/W cm⁻² and $7.25 \times 10^{-3}$ nm K⁻¹ occurred at the longest emission band at ~660 nm, corresponding to the ⁴F₉/₂ → ⁴I₁₅/₂ transition. Such wavelength tunability over several nanometers has never been demonstrated in relatively hard inorganic WGM resonators[48] and has been reported for only soft organic resonators (e.g., several tens of nanometers for polymer beads), which are sensitive to external forces such as mechanical deformation[50] or electrical fields[51]. However, soft organic materials are vulnerable to high temperature and high excitation power, which are often demanded when these materials are used as a strong light

source[22]. We found that there was no significant change in the laser linewidth depending on the temperature (<0.1 nm), which supports the excellent stability of the LQUM against thermal degradation. Our wavelength tuning over a wide spectral range was realized in a much simpler process than the wavelength tuning in semiconductor lasers that need complex external tuning devices[31].

In conclusion, an anti-Stokes-shift upconversion microlaser with an ultralow-threshold, continuous-wave operation, a narrow spectral linewidth, broad wavelength tunability, and long-term stability was demonstrated. Microscale monolithic WGM resonators were synthesized with a smooth surface texture via rapid liquid quenching of UCNPs. A theoretical analysis using atomistic simulations indirectly supports the suppression of phonon-assisted EBT in the liquid-quenched amorphous matrix. All these advantages of the LQUM enabled strong light amplification irrespective of the activator ion, resulting in low lasing thresholds. Well-defined multiwavelength outputs were observed at red, green, and blue wavelengths for $Er^{3+}$ and $Tm^{3+}$ with wavelength tunability over several nanometers. Our upconversion microlasers were directly pumped by widely commercialized 980-nm pump diode lasers, which provide a high wall-plug efficiency, a long lifetime, and applicability to industry. These excellent lasing performances of our upconversion microlasers can be further developed when they received plasmonic assistance[52]. The automated-laser-irradiation system for a microarray of self-assembled UCNPs might provide a clue for the mass-production of the LQUM upconversion microlasers. Therefore, this light source is promising for subcellular-scale biotechnologies and on-chip submicron-scale optoelectronics.

## Methods
**Materials**. $ErCl_3 \cdot 6H_2O$ (99.9%), $YCl_3 \cdot 6H_2O$ (99.9%), $YbCl_3 \cdot 6H_2O$ (99.9%), $TmCl_3 \cdot 6H_2O$ (99.9%), $HoCl_3 \cdot 6H_2O$ (99.9%), NaOH (98+%), $NH_4F$ (98+%), 1-octadecene (ODE) (90%), oleic acid (OA) (90%), Igepal CO-520, $NH_3 \cdot H_2O$ (30 wt%), and tetraethyl orthosilicate (TEOS) (99.0+%) were purchased from Sigma-Aldrich. All chemicals were used as received without further purification.

**Preparation of UCNPs**. Silica-coated $\beta$-$NaYF_4$:$Yb^{3+}$, $Er^{3+}$ (20%, 2%) UCNPs were prepared by an organometallic method. A stoichiometric amount of $LnCl_3 \cdot 6H_2O$ (1 mmol, Ln = Y, Yb, Er) was dissolved in 15 ml of ODE and 6 ml of OA under vigorous stirring in a 250-ml three-neck flask. The mixture was heated to 160 °C for 1 h under an inert atmosphere to remove water and oxygen. After cooling to 25 °C, 10 ml of methanol containing $NH_4F$ and NaOH was added dropwise and stirred for 30 min, followed by heating to 120 °C for 1 h to remove the methanol. Then, UCNPs of ~45 nm in diameter were synthesized by heating to 320 °C for 1 h. After being washed several times with ethanol, the synthesized UCNPs (1 mmol) were redispersed in 10 ml of cyclohexane. To coat a thin $SiO_2$ layer of ~8 nm in thickness, 0.4 ml of the prepared UCNP solution (0.04 mmol) was mixed with 0.5 ml of Igepal CO-520 and 9.6 ml of cyclohexane in a 20-ml glass vial and stirred for 30 min before 0.08 ml of $NH_3 \cdot H_2O$ was added. Then, 0.04 ml of TEOS, a $SiO_2$ precursor, was added to the solution, and the container was sealed hermetically with vigorous stirring for 24 h to form an amorphous silica layer. The as-prepared UCNPs coated with silica were washed with ethanol and stored in distilled water before use. To change the lanthanide activator ion from $Er^{3+}$ to $Tm^{3+}$, the procedure was identical except that $TmCl_3 \cdot 6H_2O$ was substituted for $ErCl_3 \cdot 6H_2O$.

**Liquid-quenched upconversion microspheres**. LQUMs were fabricated by following the methods in our previous work[42]. A solution of UCNPs (40 μl) was deposited onto a Si wafer (5 mm × 5 mm) to form a hemispherical drop, followed by drying at 70 °C. Then, the UCNPs were annealed at 250 °C for 12 h in a low-vacuum furnace ($7.0 \times 10^{-2}$ Torr) to facilitate laser melting. By rubbing, the UCNPs were deposited on a TEM finder grid wall and terrace. The prepared TEM finder grid was loaded on a glass slide and covered by a coverslip to allow a focused laser beam through the oil-type ×100 objective lens of an optical microscope. Under a focused laser spot of 3 MW cm$^{-2}$ (a single-mode 980-nm continuous-wave diode laser, RGB photonics, PB 980-250), the UCNPs were rapidly liquefied by the heat generated during the upconversion process and spontaneously quenched to the LQUM on the TEM finder grid wall and terrace. The detailed process is depicted in Supplementary Fig. 1 in the Supplementary Information.

**Optical characterization**. The LQUMs were characterized using the same optical setup employed for fabrication. The LQUMs were pumped and observed using the same objective lens. Before observation, the emission was filtered by a short-pass filter (95% transmittance at 400–745 nm, Chromafilter, ET750sp-2p8). The emission spectrum was measured by an EMCCD (Andor, DU970P) equipped with a spectrograph (Andor, Shamrock 303i). With a grating of 1200 lines mm$^{-1}$ (centered at 500 nm), the spectral resolution achieved by our setup was 0.12 nm (confirmed by a 532-nm laser with a linewidth of 0.28 pm). The emissive spot of the LQUM was precisely selected by an entrance slit of 20 μm to yield pure laser emission. A half-wave plate (Thorlabs, WPH10M-980) and a birefringent polarizer (Thorlabs, GL 10) were installed before and after the excitation, respectively, to investigate the transverse modes and the polarization of the laser emission.

**Microscopic temperature control**. The operation temperature of the LQUM microlasers was controlled using a home-built microscopic heating system and calibrated by an upconversion temperature sensor (Supplementary Fig. 24). Heating above 423 K was not possible due to the operating temperature range of the optical microscope. After heating, the samples were cooled naturally.

**Computational details**. An AAMD simulation, a DFT-based AIMD simulation, and a DFT calculation were performed to investigate the causes of the enhanced upconversion lasing efficiency of LQUMs. The details of the computational method and modeling are presented in the "Modeling of amorphous $NaYF_4$ + $SiO_2$ structure" and "Hexagonal crystal $NaYF_4$ structure" sections of the Supplementary Information.

**Reporting summary**. Further information on research design is available in the Nature Research Reporting Summary linked to this article.

## Data availability
The data that support the results reported in this paper and other findings of the study are available from the corresponding authors upon reasonable request.

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

## Acknowledgements

This work was financially supported by a National Research Foundation of Korea (NRF) grant funded by the Ministry of Education, Science and Technology (NRF-2019R1A6A3A0 1095776, 2019R1A2C2085177, 2020M3C1B8016137, 2020R1A5A1018052, 2014R1A5A10 09799, 2012R1A3A1050386, 2020R1A2C2102338, and 2021R1C1C2008401). The computational resources were supported by UNIST-HPC and KISTI (KSC-2020-CRE-0301).

## Author contributions

B.-S.M. performed most of the experiments, including sample preparation and optical characterizations. T.K.L. calculated the phonon properties of the materials, analyzed the upconversion mechanism, and performed the FEM simulation. W.C.J. performed modeling and atomistic simulations of the materials. S.K.K., Y.-J.K., and D.-H.K. conceived and supervised the project. All authors analyzed the data and wrote and revised the manuscript.

## Competing interests

The authors declare no competing interests.

## Additional information

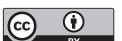

