## [Peer Review File · Nature Communications]

REVIEWER COMMENTS

Reviewer #1 (Remarks to the Author):

The manuscript entitled: "Continuous-wave upconversion lasing with a sub-10 W cm⁻² threshold enabled by atomic disorder in the host matrix", presents a low threshold and smaller upconverting microlasers than previously reported. The work is timely with the budding field of upconverting microlasers, which interest is growing fast for the potential impact on other emergent technologies. The work provides evidence of the claims and the conclusions are based on solid data collected through well-performed experiments. I can't find, either in the arguments or in the experiments any evidence of substantial flaws that could prevent its publication in Nature Communications. The work does meet the standards of the field and, in my opinion, the work could be easily reproduced in a reasonable time. After some minor revisions, my suggestion is to publish the work in Nature Communications.

Itemized comments:

Abstract:

Line 50-52: Viz. "Narrow laser lines were tuned by up to 3.56 nm by injection pump power and operation temperature adjustments." For clarity, the sentence can add that the tuning refers to spectral tuning.

Line 52, 53: Ok to leave it, but the authors may have overstated this since the plasmon nanolasers in Ref. 37 in 2019 showed thresholds competitive to the state of the art lasers.

Results and discussion:

Line 152: Viz. "it strengthens the quantum electrodynamic effect of the optical cavity." What's the meaning of that statement? It seems unrelated to the discussion up to that point, but it may make sense if they explain what's the point they want to make.

Figure 1:

I believe the FSR refers to the same mode family (polarization). So TE-TE, TM-TM. Here, the authors seem to be accounting for the separation of TE-TM.

Regarding figure organization: It looks to me that the figure could gain clarity by swapping the bottom panel in a for panels b and c (b and c can be merged into just b). It could also help in organizing the discussion.

In panel d, the circles in the inset seem to suggest that lasing is separated from the excitation/pump spot (980 nm). However, it is not intuitive to see why lasing is separated from the excitation/pump spot (980 nm). Most likely many of the Er³⁺ ions under the high excitation spot from the pump are undergoing stimulated emission.

The caption in Fig 1c reads "Because the direction of light circulation must be aligned perfectly with the circumference of the microsphere, an LQUM dangling on a TEM grid wall was selected for the formation of WGM resonators." This seems uncorrelated with panel c, which is a zoom of b.

Panel e caption. Is any of these processes simulated or are these experimental observations?

Line 180: The authors said: "LQUMs provide higher pump-to-gain interaction and lower intracavity loss than conventional upconversion microlasers".

The statement is not clear. Why is that true? I can see that smooth surfaces may reduce scattering losses but, why the LQUMs provide higher pump-to-gain interaction? Are there arguments supporting this?

Line 184: I understand that this means the pump is located at 21% of the radius from the edge (surface) but had to read it twice to get it. Maybe they can differentiate two distinct radii. R1 (center to the pump), r2 (pump to edge). I recognize this can be tricky to define, I encourage the authors to find a better way, but I understand this may be ok as it is now.

The pumping position can change several parameters at once. The reader could appreciate the authors

defining what they consider optimal pumping? Brightness, Q-factors, etc..?

About the metric used to evaluate the 'optimal' pumping. I would try to find a way of explaining the main parameters at change, either the amount of gain under the pump or optical coupling. If the effects observed are purely based on the higher gain involved, then the discussion could go around that, for the pump will excite more gain media where there is a larger overlap pump/Er³⁺ ions.

Line 195-198: Can the authors provide the slopes in a log-log plot?

Also, the statement of the lasing intensity increasing as the upconverting emission does, makes the reader think that there is no gain part in the curves, is that true? And why would that apply in this case?

Figure 2:

I don't doubt they are showing laser thresholds, but it would help to see log-log plots of the power series. I find the discussion of Figure 2 more descriptive than anything else, and the main analysis covering lasing, thresholds, and calculations are missing in the main text. Since the low thresholds are the main claim of the paper, the authors may want to make their point clear by discussing important details of how these thresholds are estimated. For example, a question to answer is why the values provided correspond to the lasing thresholds?

Line 235-237: These lines read: "Our result is notable because crystalline hexagonal NaYF₄ is the most popular host material because of its low phonon-mediated energy loss for high upconversion efficiency". I would suggest changing the wording here. Their result being notable because NaYF₄ is the most popular matrix that does not really compel much. Their result is notable for what exactly? Incorporating SiO₂ in NaYF₄, 'amorphization'? There are good arguments about this the authors can use, or they can remove the sentence and go straight to the following part.

Figure 3.

Is this figure needed in the main text? I believe it can be moved to the SI. If authors feel strongly about it, they can leave it or move it closer to Figure 1, but it diverts the attention from the lasing part IMHO. Moving this figure to the SI would make for a 5 figures paper, which may actually help to get the information across.

Line 292: The authors mentioned viz. 'Output beams'. What do they mean by beams? I find it hard to imagine why WGMs would have a preferential emission direction in the absence of surface defects or scatterers. The argument up to this point was the smooth surfaces helped in minimizing losses, hence a preferential direction in the light emission seems to contradict this very argument used previously. Do the authors mean the modes? Which seems to be what they see in Figure 4.

Figure 4:

Very interesting images in 4b. Maybe they can elaborate a bit on the significance of the images. In the caption, they mentioned high-resolution grating. Can they provide some details of the collection system in the caption? So one doesn't have to go to the methods to find it.

Figure 5:

The caption mentioned color images in panel d. I could not find the panel d, only a, b, and c.

Figure 6:

This figure is very interesting and exposes two distinct ways of tuning the mode's wavelength. The discussion seems to lack a comment about sensitivity that could be extracted from panel d, and e. And how the Q-factors may change with pump and T.

References:

There are way too many references and looks like not all are needed. For example, ref 40 is a news and views comment on some other paper, that adds no support whatsoever on any claim on this paper. Only original research will do so.

Supporting information:

Figure S2: Without having units in the intensity is hard to tell the stability.

Line 136-140: What is considered the pump-to-gain coupling distance? Can the authors define this clearly?

Figure S7: What is the coupling magnitude? How they calculated?

In principle, the spontaneous emission should be isotropic and coupling to the resonator should be symmetrical, rendering pumping schemes primarily depending on power. When the pump position is placed in the center of the spheres, the focus changes as well, changing the effective power.

I would add one more simulation, placing the upper surface of the sphere when the resonator is centered right at the pump focus.

Line 164-166: This is not needed. Rather I would like to see how things on the plots are calculated.

Figure S8: Explain how to obtain the coupling factor/value on the plots. It looks from the plots that they are just considering the pump-distance/diameter, but based on what? Intensities?

Line 168: Viz. 'Normalized pump-to-gain'; How is it calculated? Did they experimentally measure the gain or have an insight into what the effective gain might be?

Figure S9: Caption, Line 176; Viz. 'upconversion lasing (b) upconversion luminescence'; Are these nanocrystals? on a substrate, on a cavity? What's this comparison about?

It looks to me that spontaneous emission is more intense than lasing on these plots? Why? Are they plotting areas, intensities...?

Angel F. Bravo

Reviewer #2 (Remarks to the Author):

Lanthanide ions doped materials are greatly used as laser gain media due to the properties of photobleaching-resistant, nonblinking, photostable and have long lifetime real intermediate energy states for efficient population inversion. CW pumped upconversion microlasers with lower threshold attract broad interest due to its' potentials in the biological application. With high quality microcavity, remarkable lasing threshold of upconverting microlaser had been reported in the past years (Tm³⁺ doped system: 14 kW/cm², Nat. Nanotechnol. 2018, 13, 572-577; Yb³⁺/Tm³⁺ doped system: 150 W/cm², Nat. Commun. 2020 11, 6156; Yb³⁺/Er³⁺ doped system: 29 W/cm², Nat. Mater. 2019, 18, 1172-1176.). Here, Moon et al. reported an impressive record of upconversion laser with a sub-10 W/cm² threshold from an amorphous host material! The suppressed phonon-assisted energy back transfer due to the decreased phonon density of states was considered as the main reason to achieve the efficient population inversion of lanthanides ions. It is an interesting work! And to improve the quality of work toward publication, I would like to suggest a more comprehensive analysis and experimental evidence interpret their unique observation.

My major suggestions include:

1. In line 127, page 6, the description of "noticeable defects on the microsphere's surface were not observed in SEM image" sounds over-claim, as SEM is not expected to see any atomic-level defects. Here more structural characterization of the LQUM is highly recommended. For example, the elemental analysis, micro-Raman to accurately understand the inner and out-layer structures of the LQUM, as well as the phonon energy.

2. Why 2.44 μm in diameter can generate best laser emission? What's the situation for the larger size cavity? Why the smaller size cavity does not exhibit resonance with the red emission? Does this indicate that the liquid-quenching method produces the materials with different phonon energy/density and maybe also non-uniform composition or lanthanides distribution in a cavity from one to another?

3. The authors claim that the energy back transfer from the activator to the sensitizer would greatly limit the population inversion process of activator (lines 99-104). But the energy back transfer between Er³⁺ to Yb³⁺ is inessential when the doping concentration is lower than 2%. More convincing experimental evidence is expected. This can be done either by observing the suppressed EBT in high doping situation, where EBT is in high probability in NaYF₄, or quantify the ratio of energy back transfer in β-NaYF₄:Yb³⁺, Er³⁺ (20%, 2%) UCNPs and LQUM.

4. One of the advantages of lanthanide ions is the long lifetime intermediate energy states, which could promote the establishment of population inversion. It is noticeable that in the LQUM, the lifetime of each energy state will become very short, as reported in their previous paper (ref. 42). Then, how the shorted lifetime affect the threshold? How about the lifetime change below and above the threshold?

5. Clear lasing behaviours, such as the light in -light out curves in log-log plot and slope values are suggested to be provided, considering the nonlinear property of upconversion system.

6. The threshold of 6 W/cm² might be not accurate. The light in -light out curve they used is a linear one and the linewidth still keeps reducing when it reaches 1000 W/cm² (Fig. 2f). From the linewidth data, the threshold might be several kW/cm². I suggest providing more data points and reevaluate the threshold.

Minor points:

1. In line 82, page 5, the introduction of anti-Stokes-shift microlaser threshold (>103 W cm⁻²) is not accurate.

2. In line 201-203, page 10, "such narrow linewidths have never been obtained..." is over claim.

3. The authors indicate that "ETU has been difficult to incorporate into upconversion lasing" in line 157. This is not true, as ETU or EMU has been frequently employed to produce upconversion laser, see ACS Nano 11, 843-849 (2017); Nat. Commun. 7, 10304 (2016); Nat. Commun. 7, 10304 (2016), Nat. Commun. 2020 11, 6156.

4. The mechanism of energy transfer process involving multiphonon relaxation in 1e and 3f is not accurate. The threshold order might be also caused by another upconverting process, 4I_{11/2} relax to 4I_{13/2}, and then be pumped to 4F_{9/2}.

5. Why the silicon-coated hexagonal crystal NaYF₄ nanocrystal become melt at 3.16 MW/cm², while the amorphous system is stable at 3.16 MW/cm²?

6. Why the output wavelengths red-shift with the increase of pump power?

7. Why the refractive index increase with temperature increase rather than decreasing?

8. Figure 3e in lines 195 should be 3f; the first 555 in Figure 4d (coordinate axis) should be 550.

Reviewer #3 (Remarks to the Author):

In this work, by melting silica-coated upconversion nanocrystals upon high power NIR laser illumination, the high-quality microcavity for WGM lasing was fabricated. With improved surface smoothness, reduced nanoparticle scattering, and largely inhibited back-energy-transfer which supports better population inversion, WGM upconversion lasing with a low threshold was achieved. I support the publication of this work after the authors have carefully addressed the comments outlined below.

1. Authors stated that the upconversion nanocrystals are melted upon high power laser excitation in the laser-induced liquefaction process. To prove this, the authors should provide XRD data to show that the diffraction pattern was totally changed compared to the NaYF₄ host before laser treatment. Besides, High-resolution TEM images should be provided to show the disappearance of NaYF₄ nanocrystals.

2. Authors show that the energy back transfer to Yb sensitizers can be effectively eliminated by increasing the phonon-assisted relaxation of Er activators, thus improving population inversion. To support this, authors should provide Raman data for NaYF₄:Yb/Er nanocrystals without silica-coating and laser-treated microcavities.

3. Because the NIR beam can trigger laser-induced liquefaction, the sample can generate heat upon laser excitation. Authors should show lasing data of peak shifting upon a fixed excitation power for hours.

4. Authors should also provide the resonance mode simulation of prepared microcavities. A previous work on plasmon nanocavity might be a useful reference (Nature Nanotechnol. 2019, 14, 1110-1115).

5. Although this manuscript reports an interesting way of fabricating high-quantity WGM laser cavities, how

should one separate the microcavity from the TEM grid for further application? The authors should add some discussion to this issue.

REVIEWER COMMENTS

Reviewer #1 (Remarks to the Author):

The manuscript entitled: “Continuous-wave upconversion lasing with a sub-10 W cm⁻² threshold enabled by atomic disorder in the host matrix”, presents a low threshold and smaller upconverting microlasers than previously reported. The work is timely with the budding field of upconverting microlasers, which interest is growing fast for the potential impact on other emergent technologies. The work provides evidence of the claims and the conclusions are based on solid data collected through well-performed experiments. I can’t find, either in the arguments or in the experiments any evidence of substantial flaws that could prevent its publication in Nature Communications. The work does meet the standards of the field and, in my opinion, the work could be easily reproduced in a reasonable time. After some minor revisions, my suggestion is to publish the work in Nature Communications.

Itemized comments:

Abstract:

Line 50-52: Viz. “Narrow laser lines were tuned by up to 3.56 nm by injection pump power and operation temperature adjustments.” For clarity, the sentence can add that the tuning refers to spectral tuning.

Response:

We thank the reviewer for the kind correction. To avoid misunderstanding, we revised the corresponding sentence to “Narrow laser lines were spectrally tuned by up to 3.56 nm by ...” as the reviewer suggested.

Line 52, 53: Ok to leave it, but the authors may have overstated this since the plasmon

nanolasers in Ref. 37 in 2019 showed thresholds competitive to the state of the art lasers.

Response:

We thank the reviewer for the warm-hearted concern. As the reviewer pointed out, in the perspective of lasing threshold, the sentence “Our demonstration represents the first anti-Stokes-shift microlaser that is competitive against state-of-the-art Stokes-shift microlasers...” seems overstated. However, our LQUM microlaser demonstrated not only 1) low threshold ($<10 \text{ W cm}^{-2}$) but also 2) narrow linewidth ($<1 \text{ nm}$) and 3) wide spectral tunability ($\sim 3.56 \text{ nm}$), which have not been achieved together in previous upconversion-based micro/nanolasers. These properties are equally important for the performances of the microlasers; this makes us believe that the corresponding sentence is not an exaggeration. To clarify the intention of the sentence, we revised the sentence in the revised manuscript as like “Our **low-threshold, wavelength-tunable, and continuous-wave upconversion microlaser with a narrow linewidth** represents the first anti-Stokes-shift microlaser that is competitive against state-of-the-art Stokes-shift microlasers ...”

Results and discussion:

Line 152: Viz. “it strengthens the quantum electrodynamic effect of the optical cavity.” What's the meaning of that statement? It seems unrelated to the discussion up to that point, but it may make sense if they explain what's the point they want to make.

Response:

We thank the reviewer for the kind suggestion. As we agree with the reviewer's comment, to make the discussion consistent, we removed the corresponding sentence in the revised manuscript.

Figure 1:

I believe the FSR refers to the same mode family (polarization). So TE-TE, TM-TM. Here, the authors seem to be accounting for the separation of TE-TM.

Response:

We thank the reviewer for the correction. As the reviewer pointed out, the arrow was supposed to indicate the FSR of TM₂₀-TM₁₉. Therefore, we fixed this mistake in the revised manuscript.

Figure 1. Continuous-wave upconversion lasing in liquid-quenched upconversion microspheres (LQUM). a. Schematic of the fabrication of an LQUM via the liquid-quenching of silica-coated β -NaYF₄:Yb³⁺, Er³⁺ (20%, 2%) UCNP on a TEM grid using a highly focused CW 980-nm laser beam. b. SEM images of the LQUM at low and high magnifications. Because the direction of light circulation must be aligned perfectly with the circumference of the microsphere, an LQUM dangling on a TEM grid wall was selected for the formation of WGM resonators. c. Schematic of upconversion lasing in an LQUM using a free-space beam of a 980-nm continuous-wave pump laser. The LQUM accommodates upconversion lasing through the light circulation along the circumference of the microsphere upon the upconversion luminescence facilitated by energy

transfer upconversion of an activator (Er^{3+}) with a sensitizer (Yb^{3+}). d. Upconversion luminescence of UCNPs (below) and an LQUM with upconversion lasing (above). The doping concentrations of Yb^{3+} and Er^{3+} were 20% and 2%, respectively. The color inset shows an optical microscopy image of an illuminated LQUM. e. The proposed upconversion lasing pathway supported by energy transfer upconversion in the LQUM. Short-dashed, dot-dashed, colored solid, long-dashed, and dotted lines indicate absorption, energy transfer, spontaneous emission, stimulated emission, and multiphonon relaxation, respectively.

Regarding figure organization:

It looks to me that the figure could gain clarity by swapping the bottom panel in a for panels b and c (b and c can be merged into just b). It could also help in organizing the discussion.

Response:

We thank the reviewer for the kind suggestion. As we agree with the reviewer's comment, we revised the organization of **Figure 1** and its discussion in the revised manuscript.

In panel d, the circles in the inset seem to suggest that lasing is separated from the excitation/pump spot (980 nm). However, it is not intuitive to see why lasing is separated from the excitation/pump spot (980 nm). Most likely many of the Er^{3+} ions under the high excitation spot from the pump are undergoing stimulated emission.

Response:

We thank the reviewer for the critical comment. Unlike the UCNP-coated microlasers that has Er^{3+} ions only in the circumference of the WGM microresonators, the LQUM possesses Er^{3+} ions even in the inner-part of the microresonator. The inner-part Er^{3+} ions are highly advantageous for upconversion lasing because of strong pump-to-gain coupling as we addressed in **Supplementary Note 1**. However, at the pumping spot, a considerable number

of the inner-part Er^{3+} ions, which are not coupled to the WGM modes, contribute to upconversion luminescence rather than upconversion lasing as shown in **Figure 4b and 4c**. Therefore, we have to separate the observation spot from the pumping spot to accurately investigate the upconversion lasing output, which is circulating clockwise around the microresonator. To clarify our intention, we added “Because the upconversion lasing and upconversion luminescence occur simultaneously at the pumping spot, we observed the upconversion lasing at the other side away from the pumping spot to accurately investigate the lasing characteristics.” in the context, and we also revised the schematic of upconversion lasing of the LQUM in the revised manuscript (**Figure 1c**).

The caption in Fig 1c reads “Because the direction of light circulation must be aligned perfectly with the circumference of the microsphere, an LQUM dangling on a TEM grid wall was selected for the formation of WGM resonators.” This seems uncorrelated with panel c, which is a zoom of b.

Response:

We thank the reviewer for the kind concern. The corresponding sentence doesn't indicate that panel c is the exact spot of the upconversion lasing observation in panel d inset; panel c only shows that there are no noticeable surface defects on the LQUM. As we believe that panel c doesn't need to correlate with panel d inset, we left it as it is in the revised manuscript.

Panel e caption. Is any of these processes simulated or are these experimental observations?

Response:

We thank the reviewer for the comment. Panel e caption is the proposed upconversion lasing pathway which comes from both the experimental observation in **Figure 2** and the simulated

results in **Figure 3**. To avoid overstatement, we revised the caption as like “The proposed upconversion lasing pathway supported by energy transfer upconversion in the LQUM.” in the revised manuscript.

Line 180: The authors said: “LQUMs provide higher pump-to-gain interaction and lower intracavity loss than conventional upconversion microlasers”. The statement is not clear. Why is that true? I can see that smooth surfaces may reduce scattering losses but, why the LQUMs provide higher pump-to-gain interaction? Are there arguments supporting this?

Response:

We thank the reviewer for the comment. The LQUM provides higher pump-to-gain interaction compared with previously developed UCNP-coated microlasers, which has Er^{3+} ions only on the thin outer layer of the circumference because the LQUM has Er^{3+} ions in the circumference as well as the inner-part of the WGM microresonator. In the LQUM, the inner-part Er^{3+} ions can be efficiently coupled with 980 nm excitation laser beam ($2\omega_0=720$ nm for $\lambda=980$ nm, N.A.=1.3) because the pump-to-gain coupling distance is determined to be 370 nm from the surface for TM_{15} (660 nm) (FWHM of the optical mode of the WGM resonance as we defined in **Supplementary Note 1**). The high pump-to-gain interaction of the LQUM is carefully discussed in **Supplementary Note 1** and is guided right after the corresponding sentence in the manuscript, starting with “To exploit this advantage, we carefully investigated the optimal pumping position...”

Line 184: I understand that this means the pump is located at 21% of the radius from the edge (surface) but had to read it twice to get it. Maybe they can differentiate two distinct radii. R1 (center to the pump), r2 (pump to edge). I recognize this can be tricky to define, I encourage

the authors to find a better way, but I understand this may be ok as it is now.

Response:

We thank the reviewer for the kind concern. To avoid unnecessary confusion, we revised the corresponding sentence as like "...located at the rim of the LQUM (~21% of the radius far from the surface of the microsphere)." In the revised manuscript.

The pumping position can change several parameters at once. The reader could appreciate the authors defining what they consider optimal pumping? Brightness, Q-factors, etc..?

Response:

We thank the reviewer for the suggestion. In our study, the pumping position was optimized to realize the most efficient lasing (for the lowest threshold). Therefore, as the reviewer suggested, we revised the corresponding sentence as like "To exploit this advantage, we carefully investigated the optimal pumping position from the very edge to the center of an LQUM and found that the optimal pumping position for the lowest lasing threshold is located at the rim of the LQUM" in the revised manuscript.

About the metric used to evaluate the 'optimal' pumping. I would try to find a way of explaining the main parameters at change, either the amount of gain under the pump or optical coupling. If the effects observed are purely based on the higher gain involved, then the discussion could go around that, for the pump will excite more gain media where there is a larger overlap pump Er³⁺ ions.

Response:

We thank the reviewer for the comment. In **Supplementary Note 1**, we discussed the main parameters that could change upon the pumping positions; (1) heat generation in the

microcavity, (2) effective pump power, which contributes to the upconversion luminescence, and (3) the efficiency of pump-to-gain coupling. After the evaluation, we concluded that “The rim pumping provides not only a strong pump-to-gain coupling effect (vs. the center pumping) but also a low pump power loss (vs. the edge pumping), leading to the most sustainable upconversion lasing at low pump power excitation, where the heating effect is negligible.”. As we believe that the detailed discussion related to the conclusion mentioned above might distract the readers from the main context of **Figure 2**, we concisely addressed the conclusion about the optimization of the pumping positions in the main manuscript as like “To exploit this advantage, we carefully investigated the optimal pumping position from the very edge to the center of an LQUM and found that the optimal pumping position **for the lowest lasing threshold** is located at the rim of the LQUM...”

Line 195-198: Can the authors provide the slopes in a log-log plot?

Also, the statement of the lasing intensity increasing as the upconverting emission does, makes the reader think that there is no gain part in the curves, is that true? And why would that apply in this case?

Response:

We thank the reviewer for the critical comment. To provide the accurate lasing thresholds of the LQUM microlaser, we revised **Figure 2d** and **2e** from a linear plot to a log-log plot with data points at low-pump-power regions. In the revised **Figure 2d** and **2e**, we clearly show the lasing thresholds of each emission band that are determined by the slope change in a log-log plot of the emission intensity depending on pump power density, which represent the transition of the emission behavior of Er^{3+} from spontaneous emission to stimulated emission (Zhu, H., et al. "Amplified spontaneous emission and lasing from lanthanide-doped up-

conversion nanocrystals." *ACS Nano* 7(12): 11420-11426. (2013)). Furthermore, the nonlinearity in a log-log plot (the increment of the slope) indicates that the gain part exists in the upconversion lasing of the LQUM microlaser. Therefore, in the revised manuscript, we added the sentence “The lasing thresholds of each emission band were determined by the slope change in a log-log plot of the emission intensity depending on pump power density, which represents the transition of the emission behavior of Er^{3+} from spontaneous emission to stimulated emission.”

Figure 2. Ultralow-threshold continuous-wave upconversion lasing. a-b. Red- (a) and green-band (b) emission spectra of an LQUM doped with Yb^{3+} and Er^{3+} around the lasing threshold. The laser lines corresponding to ${}^2\text{H}_{11/2} \rightarrow {}^4\text{I}_{15/2}$, ${}^4\text{S}_{3/2} \rightarrow {}^4\text{I}_{15/2}$ and ${}^4\text{F}_{9/2} \rightarrow {}^4\text{I}_{15/2}$ are labeled as Peak_H, Peak_S and Peak_F, respectively. c-d. Laser emission spectrum of Peak_F near the laser threshold of sub-10 W cm^{-2} (c) and a laser emission intensity and linewidth measurement (d) e. Generation of Peak_H, and Peak_S. f. Linewidth saturation of the laser lines with increasing pump power density.

Figure 2:

I don't doubt they are showing laser thresholds, but it would help to see log-log plots of the

power series. I find the discussion of Figure 2 more descriptive than anything else, and the main analysis covering lasing, thresholds, and calculations are missing in the main text. Since the low thresholds are the main claim of the paper, the authors may want to make their point clear by discussing important details of how these thresholds are estimated. For example, a question to answer is why the values provided correspond to the lasing thresholds?

Response:

We thank the reviewer for the critical comment. In the response of the former comment, we described how we determined the upconversion lasing thresholds of the LQUM microlaser, and we revised the manuscript following the corresponding discussion.

“After the pumping position optimization, continuous-wave upconversion lasing was successfully generated at pump power densities below 10^3 W cm^{-2} (Figure 2a, b). In particular, the lowest lasing threshold of 4.7 W cm^{-2} with significant linewidth narrowing was observed at $^4\text{F}_{9/2} \rightarrow ^4\text{I}_{15/2}$ (Figure 2c, d), which is a remarkable threshold record for anti-Stokes-shift microlasers. The laser emission bands were generated following the ascending order of the energy levels of the excited states for each band, from $^4\text{F}_{9/2}$, $^4\text{S}_{3/2}$ to $^2\text{H}_{11/2}$, as shown in Figure 2d and 2e. This order indicates that multiphonon relaxation is involved in the lasing process to populate the lower-energy excited states (see the details in Figure 3f). The lasing thresholds of each emission band were determined by the slope change in a log-log plot of the emission intensity depending on pump power density, which represents the transition of the emission behavior of Er^{3+} from spontaneous emission to stimulated emission²⁵. As the pump power increases, the intensity of the laser lines increases proportionally to the upconversion luminescence intensity of the corresponding emission bands, where the power dependence of the green emission band is higher than that of the red emission band (Figure S9), thus finally yielding an intense laser spectrum, as shown in Figure 1d. Above the lasing threshold, the spectral linewidth of each lasing mode continuously decreases and converges to 0.27, 0.28, and 0.48 nm at 525, 550, and 665 nm, respectively (Figure 2f). This continuous linewidth narrowing, after the lasing threshold, along with the increasing pump power can be attributed to the increase of the degree of coherence of the stimulated emission because the number of the coherent photons that traveling around the cavity become larger at higher pump power^{35,36}.”

Line 235-237: These lines read: “Our result is notable because crystalline hexagonal NaYF₄ is the most popular host material because of its low phonon-mediated energy loss for high upconversion efficiency”. I would suggest changing the wording here. Their result being notable because NaYF₄ is the most popular matrix that does not really compel much. Their result is notable for what exactly? Incorporating SiO₂ in NaYF₄, 'amorphization'? There are good arguments about this the authors can use, or they can remove the sentence and go straight to the following part.

Response:

We thank the reviewer for the comment. As the reviewer pointed out, we concluded that the corresponding sentence is not necessary for the discussion, so we removed the corresponding sentence in the revised manuscript.

Figure 3.

Is this figure needed in the main text? I believe it can be moved to the SI. If authors feel strongly about it, they can leave it or move it closer to Figure 1, but it diverts the attention from the lasing part IMHO. Moving this figure to the SI would make for a 5 figures paper, which may actually help to get the information across.

Response:

We thank the reviewer for the kind suggestion. As the reviewer’s comment, **Figure 3** might distract the readers’ attention in the perspective of the lasing phenomena. However, in our study, the reduced phonon DOS of lanthanides in the LQUM is considered as one of the most important features for the reduction of the upconversion lasing threshold because of the

suppression of energy back transfer from Er^{3+} to Yb^{3+} , which facilitates efficient population inversion of Er^{3+} in the presence of Yb^{3+} (**Figure 3b, 3c, and 3e**). In addition, the existence of high energy phonon in the host matrix itself explains how the multiphonon relaxation contributes to the lowest lasing threshold of the lowest excited state ($^4\text{F}_{9/2}$ of Er^{3+}) (**Figure 3d and 3f**). For these reasons, we believe that **Figure 3** is needed to be in the main text, and, to guide the readers properly, **Figure 2** (the results) needs to be in front of **Figure 3** (the supporting evidence). Therefore, we would like to leave **Figure 3** as it is in the revised manuscript.

Line 292: The authors mentioned viz. ‘Output beams’. What do they mean by beams? I find it hard to imagine why WGMs would have a preferential emission direction in the absence of surface defects or scatterers. The argument up to this point was the smooth surfaces helped in minimizing losses, hence a preferential direction in the light emission seems to contradict this very argument used previously. Do the authors mean the modes? Which seems to be what they see in Figure 4.

Response:

We thank the reviewer for the critical correction. As the reviewer pointed out, the use of ‘beam’ is inappropriate in our study. The upconversion lasing in the LQUM doesn’t have a preferential emission direction. To avoid misunderstanding, we removed the terminology ‘beam’ for the upconversion lasing in the revised manuscript.

“Next, we analyzed the spatial, spectral, and polarization characteristics of the LQUM microlaser in accordance with classical laser physics, as shown in Figure 4. The **upconversion lasing outputs** are emitted from both sides of the microsphere edge...”

Figure 4:

Very interesting images in 4b. Maybe they can elaborate a bit on the significance of the images. In the caption, they mentioned high-resolution grating. Can they provide some details of the collection system in the caption? So one doesn't have to go to the methods to find it.

Response:

We thank the reviewer for the kind suggestion. As the reviewer commented, to provide the detailed experimental setup, we revised the caption of **Figure 4b** as like “Spatially resolved spectrum of laser emission from the LQUM. The narrow rectangular section was spatially selected by the spectrograph’s entrance slit of 20 μm and resolved using a high-resolution grating of 1200 lines mm^{-1} (centered at 500 nm). Along with vertical direction, each point of the selected area generates emission spectrum horizontally on the panel of EMCCD to create the image of **b**” in the revised manuscript.

Figure 5:

The caption mentioned color images in panel d. I could not find the panel d, only a, b, and c.

Response:

We thank the reviewer for the kind correction. We corrected the caption in the revised manuscript.

Figure 6:

This figure is very interesting and exposes two distinct ways of tuning the mode’s wavelength. The discussion seems to lack a comment about sensitivity that could be extracted from panel

d, and e. And how the Q-factors may change with pump and T.

Response:

We thank the reviewer for the comment. As we agree with the reviewer's comment, we revised the sentence in the context as like "Based on the fact that the lasing wavelength shift is proportional to the initial peak emission wavelength, the largest shift of 3.56 nm with the highest sensitivity of 8.76×10^{-7} nm/W cm^{-2} and 7.25×10^{-3} nm/K occurred at the longest emission band at ~660 nm, corresponding to the ${}^4F_{9/2} \rightarrow {}^4I_{15/2}$ transition."

For Q factors, we already elaborated the changes of the laser linewidth depending on the injection power in **Figure 2f**. And we also found that there was no significant change in the laser linewidth depending on the temperature (<0.1 nm), which supports the excellent stability of the LQUM against thermal degradation. We added this discussion in the context of the revised manuscript as like "We found that there was no significant change in the laser linewidth depending on the temperature (<0.1 nm), which supports the excellent stability of the LQUM against thermal degradation."

References:

There are way too many references and looks like not all are needed. For example, ref 40 is a news and views comment on some other paper, that adds no support whatsoever on any claim on this paper. Only original research will do so.

Response:

We thank the reviewer for the comment. We removed the references that are irrelevant to the conclusion in the revised manuscript.

Supporting information:

Figure S2: Without having units in the intensity is hard to tell the stability.

Response:

We thank the reviewer for the comment. We added the unit in **Figure S2** in the revised Supplementary Information.

Line 136-140: What is considered the pump-to-gain coupling distance? Can the authors define this clearly?

Response:

We thank the reviewer for the comment. The pump-to-gain coupling distance is defined as ‘the distance from the WGM resonator’s surface that covers the full-width-half-maximum of the optical fields for effective coupling of gain with pump laser beam’. We revised the sentence to clarify the definition of ‘pump-to-gain coupling distance’ in the revised **Supplementary Information Note 1** as like “Then, the effective pump-to-gain coupling distance, which is defined as the full-width-half-maximum of the optical fields from the WGM resonator’s surface for effective coupling of gain with pump laser beam, was determined to be 370 nm.”

Figure S7: What is the coupling magnitude? How they calculated?

Response:

We thank the reviewer for the comment. In our study, the pump-to-gain coupling effect is defined as the magnitude of the effective pump power that is converted into upconversion luminescence and calculated by the sum of the square of the pump intensity that overlaps with the pump-to-gain coupling distance because the upconversion at 660 nm (two photon excitation) is proportional to the pump power intensity to the second power ($\propto I^2$). Therefore,

we revised the sentence as like “Next, we defined the pump-to-gain coupling effect as the magnitude of the effective pump power that is converted into upconversion lasing. Since the upconversion at 660 nm (two photon excitation) is proportional to the pump power intensity to the second power ($\propto I^2$), we calculated the pump-to-gain coupling effect as the sum of the square of the pump intensity that overlaps with the effective pump-to-gain coupling distance.”

In principle, the spontaneous emission should be isotropic and coupling to the resonator should be symmetrical, rendering pumping schemes primarily depending on power. When the pump position is placed in the center of the spheres, the focus changes as well, changing the effective power. I would add one more simulation, placing the upper surface of the sphere when the resonator is centered right at the pump focus.

Response:

We thank the reviewer for the suggestion. As the reviewer mentioned, the spontaneous emission is isotropic and coupled to the resonator symmetrically so the simulation for focusing on the upper surface of the resonator might provide a considerable insight; the result is expected to be similar with the rim pumping because the pump laser beam has a longer optical power gradient along with the vertical direction, resulting in larger overlap than horizontal direction. However, when we pump the upper surface of the resonator, we are not able to separate the observation spot from the pumping spot because they are on a same focal plane, that hinders the experimental observation. As we concern the unsupported simulation might distract the reader, therefore, to prevent the reader from the distraction, we would like to leave **Figure S7** as it is.

Line 164-166: This is not needed. Rather I would like to see how things on the plots are

calculated.

Response:

We thank the reviewer for the kind suggestion. We removed the sentence in the revised Supplementary Information.

Figure S8: Explain how to obtain the coupling factor/value on the plots. It looks from the plots that they are just considering the pump-distance/diameter, but based on what? Intensities?

Response:

We thank the reviewer for the comment. We calculated the pump-to-gain coupling effect as we defined at **Figure S7c** (the magnitude of the effective pump power that is converted into upconversion lasing), and the magnitude was based on the sum of the square of the pump intensity that overlaps with the pump-to-gain coupling distance as we calculated in **Figure S7**.

Figure S8a shows how the magnitude of the pump-to-gain coupling effect varies depending on the diameter of the microresonator (the maximum magnitude point was normalized to 1 for fair comparison). To provide a clear guidance to the reader, we revised the caption of **Figure S8** in the revised Supplementary Information as like “**Figure S8. Pump-to-gain coupling effect as a function of microsphere size.** **a.** The normalized pump-to-gain coupling effect depending on the microsphere size. **The magnitude of the pump-to-gain coupling effect represents the effective pump power that is converted into upconversion lasing (the sum of the square of the pump intensity that overlaps with the pump-to-gain coupling distance as we calculated in Figure S7). The maximum magnitude point was normalized to 1 for fair comparison.** **b.** The maximum value of the pump-to-gain coupling effect depending on the

microsphere size. c. The relative pumping position of the maximum pump-to-gain coupling effect depending on the microsphere size.”

Line 168: Viz. ‘Normalized pump-to-gain’; How is it calculated? Did they experimentally measure the gain or have an insight into what the effective gain might be?

Response:

We thank the reviewer for the comment. Normalized pump-to-gain coupling effect was calculated as we described in **Figure S7**. **Figure S8** is not supported by experimental observation but dedicated from the simulation. Nevertheless, it provides an insight about the pump-to-gain coupling effects depending on the microresonators with various diameters.

Figure S9: Caption, Line 176; Viz. ‘upconversion lasing (b) upconversion luminescence’; Are these nanocrystals? on a substrate, on a cavity? What's this comparison about?

Response:

We thank the reviewer for the kind concern. Both the upconversion lasing and upconversion luminescence are from the LQUM at the observation spot and the pumping spot, respectively. Therefore, to avoid misunderstanding, we revised the caption in the revised Supplementary Information as like “Power dependence of the intensities and the intensity ratios of upconversion lasing **at the observation spot** (b) and upconversion luminescence **at the pumping spot** (c).”.

It looks to me that spontaneous emission is more intense than lasing on these plots? Why?

Are they plotting areas, intensities...?

Response:

We thank the reviewer for the comment. In **Figure S9b** and **9c**, the intensity was measured by the integrated intensity of the observation spot and the pumping spot for the stimulated emission (upconversion lasing) and the spontaneous emission (upconversion luminescence), respectively; as shown in **Figure 4b**, the upconversion luminescence at the pumping spot is stronger than the upconversion lasing at the observation spot as described in the previous comment. This is because why we separate the observation spot from the pumping spot for the accurate investigation of upconversion lasing in the LQUM.

Angel F. Bravo

Reviewer #2 (Remarks to the Author):

Lanthanide ions doped materials are greatly used as laser gain media due to the properties of photobleaching-resistant, nonblinking, photostable and have long lifetime real intermediate energy states for efficient population inversion. CW pumped upconversion microlasers with lower threshold attract broad interest due to its' potentials in the biological application. With high quality microcavity, remarkable lasing threshold of upconverting microlaser had been reported in the past years (Tm³⁺ doped system: 14 kW/cm², Nat. Nanotechnol. 2018, 13, 572-577; Yb³⁺/Tm³⁺ doped system: 150 W/cm², Nat. Commun. 2020 11, 6156; Yb³⁺/Er³⁺ doped system: 29 W/cm², Nat. Mater. 2019, 18, 1172-1176.). Here, Moon et al. reported an impressive record of upconversion laser with a sub-10 W/cm² threshold from an amorphous host material! The suppressed phonon-assisted energy back transfer due to the decreased phonon density of states was considered as the main reason to achieve the efficient population

inversion of lanthanides ions. It is an interesting work! And to improve the quality of work toward publication, I would like to suggest a more comprehensive analysis and experimental evidence interpret their unique observation.

My major suggestions include:

1. In line 127, page 6, the description of “noticeable defects on the microsphere’s surface were not observed in SEM image” sounds over-claim, as SEM is not expected to see any atomic-level defects. Here more structural characterization of the LQUM is highly recommended. For example, the elemental analysis, micro-Raman to accurately understand the inner and out-layer structures of the LQUM, as well as the phonon energy.

Response:

We thank the reviewer for the comment. To provide more convincing structural characterizations, we added the Raman spectrum of β -NaYF₄ and the LQUM as shown in **Figure S20** (newly added) in the revised Supplementary Information. The decreased Raman intensity of the LQUM compared with β -NaYF₄ successfully supports the greatly reduced phonon DOS of the LQUM (**Figure 3b, 3c and 3d**)

Figure S20. Raman spectrum of β -NaYF₄, LQUM, and nickel TEM grid (background). To avoid unnecessary excitation of excited states of Er³⁺, we employed 785 nm laser for Raman spectroscopy (532 nm excites ⁴S_{3/2}, and 633 nm excites ⁴F_{9/2}). For a fair comparison, we investigated each sample on nickel TEM grid and normalize the Raman intensities by sample volume.

As the reviewer concerns, SEM is not appropriate for the atomic-level defect analysis. However, in the case of dielectric WGM microresonators, SEM is considered sufficient for the defect analysis on the out-layer microstructure of the microresonators because the defects smaller than several nanometers doesn't limit the quality factor of the microresonators; it doesn't act as a scattering center for the scattering loss (Righini, G., et al. "Whispering gallery mode microresonators: fundamentals and applications." *Riv. del Nuovo Cim.* 34(7): 435-488. (2011), He, L., et al. "Whispering gallery microcavity lasers." *Laser Photonics Rev.* 7(1): 60-82. (2013)). In addition, we already reported the inner microstructure of the liquid-quenched amorphous materials in our previously paper (Moon, B. S., et al. "Ultrafast Single-Band Upconversion Luminescence in a Liquid-Quenched Amorphous Matrix." *Adv. Mater.* 30(25): e1800008 (2018)) and showed the complete monolithic integration of the elements upon the liquid-quenching process. From these reasons, we did not add detailed further microstructural analysis on the inner and out-layer of the LQUM in the present manuscript. To provide a clear guidance for the reader as suggested by the reviewer, we revised the sentence as like "The molten upconversion host matrix was irreversibly solidified into a uniform and smooth microsphere by surface tension during liquefaction as shown in Figure 1b (the microstructural analysis of the liquid-quenched amorphous materials is provided in our previous report³⁰). Because the surface tension results in excellent surface finish¹¹, noticeable defects for the scattering center were not observed on the microsphere's surface in a high-resolution (1.4 nm) SEM image."

2. Why 2.44 μm in diameter can generate best laser emission? What's the situation for the larger size cavity? Why the smaller size cavity does not exhibit resonance with the red

emission? Does this indicate that the liquid-quenching method produces the materials with different phonon energy/density and maybe also non-uniform composition or lanthanides distribution in a cavity from one to another?

Response:

We thank the reviewer for the comment. As we demonstrated in our previous paper (Moon, B. S., et al. "Ultrafast Single-Band Upconversion Luminescence in a Liquid-Quenched Amorphous Matrix." *Adv. Mater.* 30(25): e1800008 (2018)), there was a difficulty in fabricating the LQUM larger than the laser spot size ($2\ \mu\text{m} \times 4\ \mu\text{m}$, an elliptical shape originated from the astigmatism of diode lasers) because UCNPs out of the laser spot cannot participate in the liquid-quenching process. However, we would like to emphasize that the general goal of the microlaser research field is to miniaturize a microresonator. To provide clearer information, we added this discussion in the context of the revised manuscript as like "Note that the LQUM larger than the laser spot size ($2\ \mu\text{m} \times 4\ \mu\text{m}$, an elliptical shape originated from the astigmatism of diode lasers) is hardly fabricated because UCNPs out of the laser spot cannot participate in the liquid-quenching process.". On the other hand, the smaller size cavity does not exhibit resonance with the red emission because the positions of resonance modes are not matched with the red emission band. We believe that the absence of red emission in the smaller LQUM microlasers is not from the variations in the fabrication of the LQUM, which is supported by the identical emission spectrum of the upconversion luminescence from the LQUM with various sizes as shown in **Figure 1d and S4**.

3. The authors claim that the energy back transfer from the activator to the sensitizer would greatly limit the population inversion process of activator (lines 99-104). But the energy back transfer between Er^{3+} to Yb^{3+} is inessential when the doping concentration is lower than 2%.

More convincing experimental evidence is expected. This can be done either by observing the suppressed EBT in high doping situation, where EBT is in high probability in NaYF₄, or quantify the ratio of energy back transfer in β -NaYF₄:Yb³⁺, Er³⁺ (20%, 2%) UCNPs and LQUM.

[redacted]

Meanwhile, the dramatically reduced phonon DOS in the LQUM (the simulation of Figure 3) is confirmed by comparing the Raman spectrum of β -NaYF₄ and the LQUM in Figure S20 (newly added in the revised Supplementary Information). To provide the supporting evidence for the conclusion, we added the sentences “Note that the simulated phonon energy of the LQUM is supported by the greatly reduced Raman emission intensity of the LQUM compared with β -NaYF₄ (Figure S20).” in the context of the revised manuscript.

Figure S20. Raman spectrum of β -NaYF₄, LQUM, and nickel TEM grid (background). To avoid unnecessary excitation of excited states of Er³⁺, we employed 785 nm laser for Raman spectroscopy (532 nm excites ⁴S_{3/2}, and 633 nm excites ⁴F_{9/2}). For a fair comparison, we investigated each sample on nickel TEM grid and normalize the Raman intensities by sample volume.

4. One of the advantages of lanthanide ions is the long lifetime intermediate energy states, which could promote the establishment of population inversion. It is noticeable that in the LQUM, the lifetime of each energy state will become very short, as reported in their previous paper (ref. 42). Then, how the shorted lifetime affect the threshold? How about the lifetime change below and above the threshold?

Response:

We thank the reviewer for the comment. As the reviewer pointed out, the long-lived intermediate states of lanthanide are considered advantageous for the population inversion. However, in classical lasing physics, the lasing threshold varies less depending on the lifetime of the excited states because the long lifetime is inversely proportional to the emission cross section; the lasing threshold depends on the product of the lifetime and the emission cross section of the excited states, which is originated from the material itself (D. S. Sumida and T. Y. Fan, “Effect of radiation trapping on fluorescence lifetime and emission cross section measurements in solid-state laser media”, *Opt. Lett.* 19 (17), 1343 (1994), H. Kühn et al., “Model for the calculation of radiation trapping and description of the pinhole method”, *Opt. Lett.* 32 (13), 1908 (2007), I. G. Kisialiou, “Free of reabsorption upper-state lifetime measurements by the method of transient gratings”, *Appl. Opt.* 51 (22), 5458 (2012)). Therefore, we concluded that the low lasing threshold observed in the LQUM is originated from the suppression of EBT due to the dramatically reduced phonon DOS of lanthanides rather than the shortened lifetime of the excited states. Note that, for the lifetime changes below and above the lasing threshold, we could not measure the lifetime of the upconversion

lasing because the lasing threshold is too low ($<10 \text{ W cm}^{-2}$) to generate a sufficient signal for time-resolved measurement.

5. Clear lasing behaviours, such as the light in -light out curves in log-log plot and slope values are suggested to be provided, considering the nonlinear property of upconversion system.

Response:

We thank the reviewer for the critical comment. To provide the accurate lasing thresholds of the LQUM microlaser, we revised **Figure 2d** and **2e** from a linear plot to a log-log plot with data points at low-pump-power regions. In the revised **Figure 2d** and **2e**, we clearly show the lasing thresholds of each emission band which are determined by the slope change in a log-log plot of the emission intensity depending on pump power density, which represents the transition of the emission behavior of Er^{3+} from spontaneous emission to stimulated emission (Zhu, H., et al. "Amplified spontaneous emission and lasing from lanthanide-doped up-conversion nanocrystals." *ACS Nano* 7(12): 11420-11426. (2013)). Furthermore, the nonlinearity in a log-log plot (the increment of the slop) indicates that the gain part exists in the upconversion lasing of the LQUM microlaser. Therefore, in the revised manuscript, we added the sentence "The lasing thresholds of each emission band were determined by the slope change in a log-log plot of the emission intensity depending on pump power density, which represents the transition of the emission behavior of Er^{3+} from spontaneous emission to stimulated emission²⁵."

Figure 2. Ultralow-threshold continuous-wave upconversion lasing. a-b. Red- (a) and green-band (b) emission spectra of an LQUM doped with Yb^{3+} and Er^{3+} around the lasing threshold. The laser lines corresponding to ${}^2\text{H}_{11/2} \rightarrow {}^4\text{I}_{15/2}$, ${}^4\text{S}_{3/2} \rightarrow {}^4\text{I}_{15/2}$ and ${}^4\text{F}_{9/2} \rightarrow {}^4\text{I}_{15/2}$ are labeled as Peak_H, Peak_S and Peak_F, respectively. c-d. Laser emission spectrum of Peak_F near the laser threshold of sub-10 W cm^{-2} (c) and a laser emission intensity and linewidth measurement (d). e. Generation of Peak_H, and Peak_S. f. Linewidth saturation of the laser lines with increasing pump power density.

6. The threshold of 6 W/cm^2 might be not accurate. The light in -light out curve they used is a linear one and the linewidth still keeps reducing when it reaches 1000 W/cm^2 (Fig. 2f). From the linewidth data, the threshold might be several kW/cm^2 . I suggest providing more data points and reevaluate the threshold.

Response:

We thank the reviewer for the critical comment. After we accurately determined the upconversion lasing thresholds of the LQUM microlaser by the slope change in a log-log pot, we found that the lowest lasing threshold is 4.7 W cm^{-2} for ${}^4\text{F}_{9/2} \rightarrow {}^4\text{I}_{15/2}$. Furthermore, we show the significant linewidth narrowing around the lasing threshold in **Figure 2d**. On the other hand, as the reviewer pointed out, we also observed continuous linewidth narrowing

from the lasing thresholds to several kW cm⁻² at each emission band, $^2H_{11/2} \rightarrow ^4I_{15/2}$, $^4S_{3/2} \rightarrow ^4I_{15/2}$ and $^4F_{9/2} \rightarrow ^4I_{15/2}$. This continuous linewidth narrowing, after the lasing threshold, along with the increasing pump power can be attributed to the increase of the degree of coherence of the stimulated emission because the number of the coherent photons that traveling around the cavity become larger at higher pump power (Strauf, S. et al. Self-tuned quantum dot gain in photonic crystal lasers. Phys. Rev. Lett. 96, 127404 (2006). Khajavikhan, M. et al. Thresholdless nanoscale coaxial lasers. Nature 482, 204-207 (2012)). Therefore, to guide the reader properly, we added the discussion above in the context of the revised manuscript as like “This continuous linewidth narrowing, after the lasing threshold, along with the increasing pump power can be attributed to the increase of the degree of coherence of the stimulated emission because the number of the coherent photons that traveling around the cavity become larger at higher pump power^{35,36}”.

Minor points:

1. In line 82, page 5, the introduction of anti-Stokes-shift microlaser threshold ($>10^3$ W cm⁻²) is not accurate.

Response:

We thank the reviewer for the kind correction. In the introduction, for the fair comparison of the performances between the microlasers, we intended to distinguish the dielectric-cavity-based microlasers from the plasmonic-cavity-based nanolasers because the plasmonic modes of the plasmonic-cavity-based nanolasers confine photons within sub-wavelength regions to facilitate the effects of cavity quantum electrodynamics (QED) (*e.g.*, the enhancement of the spontaneous emission rate (Purcell effect) and a large spontaneous emission coupling to the

lasing mode (β -factor)) for the reduction of lasing thresholds at the expense of quality factor due to the propagation loss of metallic medium (Khajavikhan, M., et al. "Thresholdless nanoscale coaxial lasers." *Nature* 482 (7384): 204-207. (2012), Hayenga, W. E., et al. "Second-order coherence properties of metallic nanolasers." *Optica* 3(11) (2016) Hill, M. T. and M. C. Gather. "Advances in small lasers." *Nat. Photonics* 8 (12): 908-918. (2014)). To clarify our intention, we added "dielectric-cavity-based" in the context. In addition, since the latest reference, which demonstrated dielectric-cavity-based upconversion lasing threshold as low as 150 W cm^{-2} (Shang, Y. et al. Low threshold lasing emissions from a single upconversion nanocrystal. *Nat. Commun.* 11, 6156 (2020)), has been published, we corrected the corresponding sentence with adding the reference as like "Therefore, for the past decade, a number of attempts have been made to realize anti-Stokes-shift microlasers using UCNPs, mainly by coating UCNPs on the surfaces of spherical or cylindrical dielectric-cavity-based microresonators^{22, 24-27}. These microlasers typically involve a lasing threshold ($>10^2 \text{ W cm}^{-2}$) that is two orders of magnitude higher than that associated with Stokes-shift laser operation ($<10^1 \text{ W cm}^{-2}$)^{6,28} due to the high pump power requirement for multiphoton absorption."

2. In line 201-203, page 10, "such narrow linewidths have never been obtained..." is over claim.

Response:

We thank the reviewer for the kind correction. As we agree with the reviewer, we removed that part in the revised manuscript.

3. The authors indicate that "ETU has been difficult to incorporate into upconversion lasing" in line 157. This is not true, as ETU or EMU has been frequently employed to produce

upconversion laser, see ACS Nano 11, 843-849 (2017); Nat. Commun. 7, 10304 (2016); Nat. Commun. 7, 10304 (2016), Nat. Commun. 2020 11, 6156.

Response:

We thank the reviewer for the kind concern. Although there are a number of papers that demonstrate upconversion lasing in the crystalline host materials (*e.g.*, β -NaYF₄) with ETU system as the reviewer mentioned, the demonstrations were realized by a pulsed excitation scheme, which provides high peak pump power. With high peak pump power, we believe that energy back transfer (EBT) is not a limiting factor for the establishment of population inversion because the pump power is sufficient to overcome the effects of EBT at the moment of the excitation. However, with low peak pump power of continuous-wave operation, EBT can be the limiting factor because the significant depopulation of the excited state of Er³⁺ occurs as described in the literature (Scheeps, R.. "Upconversion laser processes." *Progress in Quantum Electronics* 20(4): 271-358. (1996)). For that reason, there has not been a demonstration of continuous-wave upconversion microlaser for the crystalline host materials (*e.g.*, β -NaYF₄) with ETU system except for the recent demonstration that exploits greatly-enhanced electric fields of a well-designed lattice plasmon cavity (Fernandez-Bravo, A., et al. "Ultralow-threshold, continuous-wave upconverting lasing from subwavelength plasmons." *Nat. Mater.* 18(11): 1172-1176. (2019))

Therefore, to clarify the discussion above, we revised the sentence in the context of the revised manuscript as like "Nevertheless, because the use of sensitizers generally hinders the population inversion of activators (*e.g.*, Er³⁺ or Tm³⁺) through detrimental energy back transfer (EBT) to sensitizers, ETU has been difficult to incorporate into continuous-wave upconversion microlasers."

4. The mechanism of energy transfer process involving multiphonon relaxation in 1e and 3f is not accurate. The threshold order might be also caused by another upconverting process, $4I_{11/2}$ relax to $4I_{13/2}$, and then be pumped to $4F_{9/2}$.

Response:

We thank the reviewer for the kind suggestion. As the reviewer suggested, the upconversion luminescence of the conventional upconversion materials (*e.g.*, β -NaYF₄:Yb³⁺, Er³⁺) can be triggered by the route of Er³⁺: $4I_{11/2} \rightarrow 4I_{13/2} \rightarrow 4F_{9/2}$ (multiphonon relaxation, followed by energy transfer from Yb³⁺). However, in the case of the LQUM, the energy transfer from Yb³⁺ to Er³⁺ for the excitation of $4I_{13/2} \rightarrow 4F_{9/2}$ is expected to be hardly takes place because of the large energy gap between two transitions (Er³⁺: $4I_{13/2} \rightarrow 4F_{9/2}$ and Yb³⁺: $2F_{5/2} \rightarrow 2F_{7/2}$, $\sim 1500 \text{ cm}^{-1}$); the dramatically reduced phonon DOS of lanthanides in the LQUM effectively prevents phonon-assisted energy transfers between Er³⁺ and Yb³⁺ and only multiphonon relaxation will excite $4F_{9/2}$ of Er³⁺ to emit red emission in the LQUM. Therefore, we left **Figure 1e** and **Figure 3f** as it is in the revised manuscript.

5. Why the silicon-coated hexagonal crystal NaYF₄ nanocrystal become melt at 3.16 MW/cm², while the amorphous system is stable at 3.16 MW/cm²?

Response:

We thank the reviewer for the comment. As we described in our previously published paper (Moon, B. S., et al. "Ultrafast Single-Band Upconversion Luminescence in a Liquid-Quenched Amorphous Matrix." *Adv. Mater.* 30(25): e1800008 (2018)), the principles for the liquid-quenching process of upconversion nanoparticles are like below:

“The crumb of the nanoparticles are readily melted upon highly focused laser beam because of the nanoparticles’ surface effects; 1) the reduced melting point of nanoparticles compared with the bulk (Lindemann criterion), 2) the accelerated heating in the crumb of nanoparticles (the nanoscale confinement of the phonon propagation). Then, the molten nanoparticles are integrated monolithically due to the surface tension effect. After the integration, the material becomes thermally stable because of the extinction of the nanoparticles’ surface effects.”. We believe that this description would be the answer for the question.

6. Why the output wavelengths red-shift with the increase of pump power?

Response:

We thank the reviewer’s comment. The red-shift upon the increase of pump power is attributed to the increased cavity circumference, because of thermal expansion, that is responsible for the increased wavelength of the resonance modes. We described this in the manuscript as like “The increase in the internal temperature of the LQUM microlaser causes thermal expansion of the cavity which induces a red-shift of the lasing wavelengths.”

7. Why the refractive index increase with temperature increase rather than decreasing?

Response:

We thank the reviewer’s kind correction. The refractive index of the LQUM doesn’t increase with temperature. We sincerely apologize that mistake. We removed that discussion in the revised manuscript.

8. Figure 3e in lines 195 should be 3f;

Response:

We thank the reviewer's kind correction. We corrected this typo in the revised manuscript.

the first 555 in Figure 4d (coordinate axis) should be 550.

Response:

We thank the reviewer's kind correction. We corrected the x-axis value in the revised manuscript.

Reviewer #3 (Remarks to the Author):

In this work, by melting silica-coated upconversion nanocrystals upon high power NIR laser illumination, the high-quality microcavity for WGM lasing was fabricated. With improved surface smoothness, reduced nanoparticle scattering, and largely inhibited back-energy-transfer which supports better population inversion, WGM upconversion lasing with a low threshold was achieved. I support the publication of this work after the authors have carefully addressed the comments outlined below.

1. Authors stated that the upconversion nanocrystals are melted upon high power laser excitation in the laser-induced liquefaction process. To prove this, the authors should provide XRD data to show that the diffraction pattern was totally changed compared to the NaYF₄ host before laser treatment. Besides, High-resolution TEM images should be provided to show the disappearance of NaYF₄ nanocrystals.

Response:

We thank the reviewer for the comment. The microstructural analysis (including HRTEM images, SAED patterns and EDS analysis) of the liquid-quenched amorphous materials compared with hexagonal NaYF₄ crystals was showed in our previously published paper (Moon, B. S., et al. "Ultrafast Single-Band Upconversion Luminescence in a Liquid-Quenched Amorphous Matrix." *Adv. Mater.* 30(25): e1800008 (2018)) to prove the complete mixing and integration of the elements upon the liquid-quenching process. Therefore, to provide a clear guidance for the reader, we added "the microstructural analysis of the liquid-quenched amorphous materials is provided in our previous report" in the context of **Figure 1**, which describes the fabrication of the liquid-quenched amorphous materials.

2. Authors show that the energy back transfer to Yb sensitizers can be effectively eliminated by increasing the phonon-assisted relaxation of Er activators, thus improving population inversion. To support this, authors should provide Raman data for NaYF₄:Yb/Er nanocrystals without silica-coating and laser-treated microcavities.

Response:

We thank the reviewer for the critical comment. As the reviewer pointed out, the comparison of Raman spectrum between β -NaYF₄ and the LQUM could support the dramatically reduced phonon DOS in the LQUM so we added Raman spectrum of each host materials (**Figure S20** in the revised **Supplementary Information**).

Figure S20. Raman spectrum of β -NaYF₄, LQUM, and nickel TEM grid (background). To avoid unnecessary excitation of excited states of Er³⁺, we employed 785 nm laser for Raman spectroscopy (532 nm excites ⁴S_{3/2}, and 633 nm excites ⁴F_{9/2}). For a fair comparison, we investigated each sample on nickel TEM grid and normalize the Raman intensities by sample volume.

To provide the supporting evidence for the conclusion, we added the sentences “**Note that the simulated phonon energy of the LQUM is supported by the greatly reduced Raman emission intensity of the LQUM compared with β -NaYF₄ (Figure S20).**” in the context of the revised manuscript.

3. Because the NIR beam can trigger laser-induced liquefaction, the sample can generate heat upon laser excitation. Authors should show lasing data of peak shifting upon a fixed excitation power for hours.

Response:

We thank the reviewer’s kind concern. In **Figure 6c** (upper panel), we added the peak shifting data upon a fixed excitation power and temperature for hours.

Figure 6. Tuning of laser emission wavelengths. **a.** Schematic of wavelength tuning achieved by controlling the pump power density (ΔP) and operation temperature (ΔT). The initial lasing wavelength and the shifted lasing wavelengths under the control of the pump power density and operation temperature are indicated by λ_0 , λ_p and λ_T , respectively. **b.** Emission spectra of spectrally tuned lasers. $\Delta\lambda_p$ and $\Delta\lambda_T$ indicate shifts in the lasing wavelength with increasing pump power density (from 0.01 to 3.16 MW cm^{-2}) and operation temperature (from 298 to 423 K). **c.** Stability analysis in terms of static (above) and dynamic (below) operations. Green and purple lines in static operation indicate the lasing peak intensity and spectral position depending on time, respectively. **d-e.** Linear plots of the laser line peak wavelengths as a function of pump power density (**d**) and operation temperature (**e**).

4. Authors should also provide the resonance mode simulation of prepared microcavities. A previous work on plasmon nanocavity might be a useful reference (Nature Nanotechnol. 2019, 14, 1110–1115).

We thank the reviewer for the fruitful suggestion. As we agree with the reviewer's comment, we performed the finite element method (FEM) based simulation to investigate the whispering gallery mode (WGM) of our upconversion microsphere (Figure S22a). In Figure

S22b, we found that the WGMs of our upconversion microsphere with transverse magnetic (TM) modes (*i.e.*, TM_{20} , TM_{19} , and TM_{15}) were clearly shown around the intense lasing emission wavelengths (*i.e.*, 525 nm, 550 nm, and 665 nm). Accordingly, due to the matching with the WGMs and upconverted emission wavelengths, our upconversion microlaser can be well activated.

Figure S22. WGM simulation of upconversion microsphere. **a.** Model system for WGM simulation of upconversion microsphere. 2D axisymmetric simulation is performed. Diameter of upconversion microsphere (D_{UC}), thickness of air region (T_{air}), thickness of perfect matched layer (T_{PML}), and length of system (L_{system}) are 2.44, 5, 1, and 10 μm , respectively. **b.** Electric field distributions of upconversion microsphere. Two types of distribution are shown; 3D and 2D cross section distributions. Note that red colored arrows indicate the directions of magnetic field, which describe the transverse magnetic (TM) modes.

Therefore, we added the computational detail of FEM based simulation and the simulated results in **Supplementary Note 3** and **Figure S22** of Supplementary Information and added the sentence “The finite element method (FEM) based simulations were also supported the WGM of LQUM (Supplementary Note 3 and Figure S22)” in the revised manuscript.

Supplementary Note 3

Computational details

Investigation of WGM in upconversion microsphere

To investigate the WGM of our upconversion microsphere, we performed finite element method (FEM) based simulation. In this simulation, the two dimensional (2D) axisymmetric and the wave optics module (electromagnetic waves and frequency domain) are used. The 2D axisymmetric method is a quasi-three dimensional simulation with a certain azimuthal mode number. The extremely fine setting of mesh is adopted with electromagnetic waves and frequency domain contributor. The azimuthal mode numbers are set as 20, 19, and 15 for 525 nm, 550 nm, and 665 nm lasing emission wavelengths, respectively. The refractive index of upconversion microsphere is obtained from the **Figure S21**. The ambient medium is air with 5 μm thickness. The perfect matched layer (PML) is adopted with 1 μm thickness. The refractive index of PML is also air condition. The size of upconversion microsphere is 2.44 μm . The detailed model description is in **Figure S22a**. All simulations were performed using COMSOL Multiphysics software (ver. 5.4).

Figure S22. WGM simulation of upconversion microsphere. **a.** Model system for WGM simulation of upconversion microsphere. 2D axisymmetric simulation is performed. Diameter of upconversion microsphere (D_{UC}), thickness of air region (T_{air}), thickness of perfect matched layer (T_{PML}), and length of system (L_{system}) are 2.44, 5, 1, and 10 μm , respectively. **b.** Electric field distributions of upconversion microsphere. Two types of distribution are shown; 3D and 2D cross section distributions. Note that red colored arrows indicate the directions of magnetic field, which describe the transverse magnetic (TM) modes.

To investigate the WGM of our upconversion microsphere, we performed the FEM based simulation (**Supplementary Note 3** and **Figure S22a**). In **Figure S22b**, we found that the WGMs of our upconversion microsphere with TM modes (*i.e.*, TM_{20} , TM_{19} , and TM_{15}) were clearly shown around the intense lasing emission wavelengths (*i.e.*, 525 nm, 550 nm, and 665 nm). Accordingly, due to the matching with the WGMs and upconverted emission wavelengths, our upconversion microlaser can be well activated.

5. Although this manuscript reports an interesting way of fabricating high-quantity WGM

laser cavities, how should one separate the microcavity from the TEM grid for further application? The authors should add some discussion to this issue.

Response:

We thank the reviewer's critical comment. We are currently developing a method for the mass-production of the LQUM upconversion microlasers using the automated-laser-irradiation system and a microarray of self-assembled upconversion nanoparticles as precursors. This new fabrication system will be reported a separate manuscript within this year. Per the reviewer's suggestion, we added a sentence in Conclusion as like "The automated-laser-irradiation system for a microarray of self-assembled upconversion nanoparticles might provide a clue for the mass-production of the LQUM upconversion microlasers."

REVIEWERS' COMMENTS

Reviewer #1 (Remarks to the Author):

Although some of the questions raised were only partially addressed, e. g. the authors didn't address anywhere why they observed slopes of 9 in the green band but only 2 in the red band, the authors did a good job integrating the comments from reviewers into the main text.

There is some ambiguity defining the coupling distance as the FWHM of the optical fields, doesn't seem intuitive for the general audience. The paper could benefit from a more detailed or intuitive description. Overall, the manuscript looks quite polished now and it touches on important aspects of the upconverting lasing that will be useful for other research groups in the field.

Reviewer #2 (Remarks to the Author):

The authors have answered in a satisfying manner to the issues raised and I recommend publication.

Reviewer #3 (Remarks to the Author):

The authors have addressed my concerns in this revised manuscript. I recommend its publication.

REVIEWERS' COMMENTS

Reviewer #1 (Remarks to the Author):

Although some of the questions raised were only partially addressed, e. g. the authors didn't address anywhere why they observed slopes of 9 in the green band but only 2 in the red band, the authors did a good job integrating the comments from reviewers into the main text.

There is some ambiguity defining the coupling distance as the FWHM of the optical fields, doesn't seem intuitive for the general audience. The paper could benefit from a more detailed or intuitive description.

Overall, the manuscript looks quite polished now and it touches on important aspects of the upconverting lasing that will be useful for other research groups in the field.

Response: We heartly thank the reviewer for kind support for the publication. To address a reason of the different slopes in green and red emission bands, we added the sentence “The higher slope values of green emission bands (${}^4F_{9/2}$, ${}^4S_{3/2} \rightarrow {}^4I_{15/2}$) than red emission band (${}^4F_{9/2} \rightarrow {}^4I_{15/2}$) at lasing thresholds might be attributed to the higher amount of input photons at green emission bands than red emission band (see the upconversion luminescence of the LQUM that has intense single-band emission at green region as shown in Figure 1d).” in the context of the revised manuscript. Furthermore, as we newly defined the coupling distance in our manuscript to quantify the pump-to-gain interaction, we revised the sentence as “To quantify the pump-to-gain interaction, we newly defined the effective pump-to-gain coupling distance, which is defined as the full-width-half-maximum of the optical fields from the WGM resonator's surface for effective coupling of gain with pump laser beam, was determined to be 370 nm.” in the revised Supplementary Information.

Reviewer #2 (Remarks to the Author):

The authors have answered in a satisfying manner to the issues raised and I recommend publication.

Response: We heartly thank the reviewer for kind support for the publication.

Reviewer #3 (Remarks to the Author):

The authors have addressed my concerns in this revised manuscript. I recommend its publication.

Response: We heartly thank the reviewer for kind support for the publication.